# DriveCamSim: Generalizable Camera Simulation via Explicit Camera Modeling for Autonomous Driving

## Abstract

Camera sensor simulation serves as a critical role for autonomous driving (AD), e.g. evaluating vision-based AD algorithms. While existing approaches have leveraged generative models for controllable image/video generation, they remain constrained to generating multi-view video sequences with fixed camera viewpoints and video frequency, significantly limiting their downstream applications. To address this, we present a generalizable camera simulation framework Drive-CamSim, whose core innovation lies in the proposed Explicit Camera Modeling (ECM) mechanism. Instead of implicit interaction through vanilla attention, ECM establishes explicit pixel-wise correspondences across multi-view and multi-frame dimensions, decoupling the model from overfitting to the specific camera configurations (intrinsic/extrinsic parameters, number of views) and temporal sampling rates presented in the training data. For controllable generation, we identify the issue of information loss inherent in existing conditional encoding and injection pipelines, proposing an information-preserving control mechanism. This control mechanism not only improves conditional controllability, but also can be extended to be identity-aware to enhance temporal consistency in foreground object rendering. With above designs, our model demonstrates superior performance in both visual quality and controllability, as well as generalization capability across spatial-level (camera parameters variations) and temporal-level (video frame rate variations), enabling flexible user-customizable camera simulation tailored to diverse application scenarios.

## 1 Introduction

The field of autonomous driving (AD) has witnessed significant progress in recent years, benefiting from the emergence of large-scale datasets and technological progress. This evolution has propelled the paradigm shift from conventional modular frameworks to integrated end-to-end systems (Hu et al., 2023; Jiang et al., 2023; Sun et al., 2024; Liao et al., 2024) and knowledge-enhanced learning methodologies (Tian et al., 2024; Wen et al., 2023; Jiang et al., 2024a). Despite demonstrating impressive performance on standardized benchmarks, critical limitations persist in terms of generalization capability and performance in corner cases. These shortcomings primarily stem from the limited data diversity inherent in existing evaluation frameworks, highlighting the urgent need for more realistic simulation platforms.

To advance the development of vision-based autonomous driving (AD) algorithms, recent studies have leveraged advanced techniques for synthesizing multi-view driving scenes. Among these, two representative technical approaches are rendering-based methods and generative models, each exhibiting distinct advantages and limitations.

Rendering-based techniques, such as NeRF (Mildenhall et al., 2021) and 3D Gaussian Splatting (Kerbl et al., 2023), excel at maintaining high consistency in novel view synthesis. However, they typically require per-scene optimization, which limits their ability to benefit from scaling laws of data and computation. Furthermore, these methods often suffer from significant degradation in rendering quality on novel trajectories with large lateral displacements, due to the sparse observation and low reconstruction quality in driving scenes.

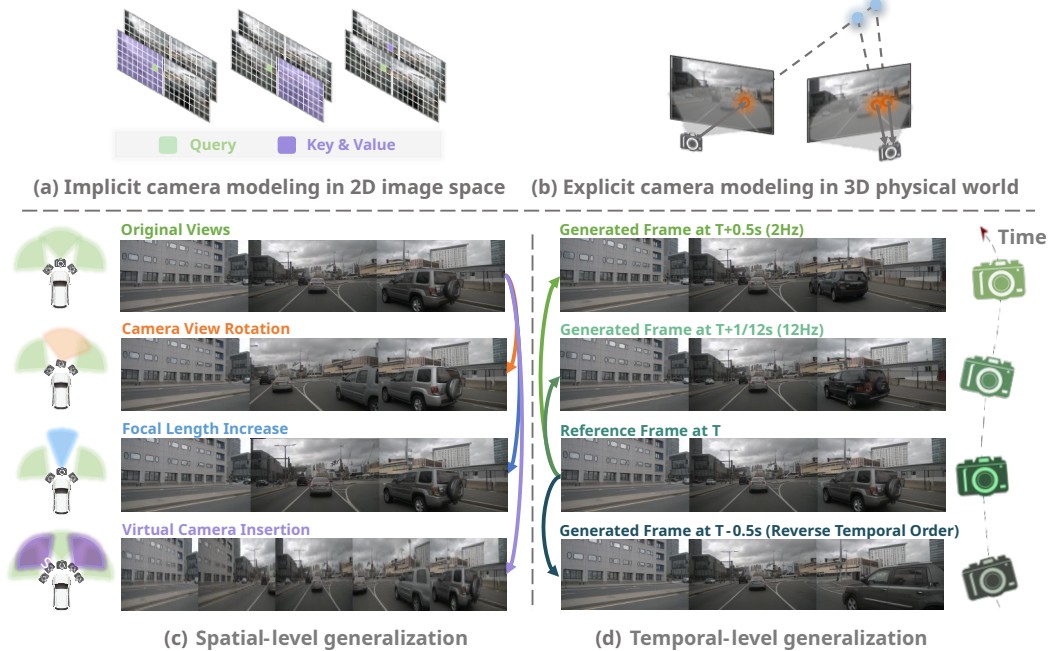

Figure 1: Instead of (a) implicit camera modeling in 2D image space, we propose (b) explicit camera modeling in 3D physical world to unleash the (c) spatial-level and (d) temporal-level generalization capabilities for flexible camera simulation.

The generative models, such as diffusion models (Ho et al., 2020), can achieve improved performance with increasing data and computation, exhibiting a higher performance ceiling, but exhibit poor generalization capability across spatial-level (camera parameters variations) and temporal-level (video frame rate variations). The reason behind this is that prior works inherently assume fixed camera parameters and frame rates, typically employing vanilla attention to implicitly model cross-view and cross-frame interactions, as shown in Fig 1 (a). This can be seen as an implicit camera modeling in 2D image space that overfits to specific camera parameters and video frequency presented in training dataset, thus exhibit poor generalization capability, severely restricting their practical applications.

A natural question arises that can we integrate the strengths of both approaches while mitigating their respective limitations? Motivated by this, we propose DriveCamSim, a generalizable generative camera simulation framework with the core lying in Explicit Camera Modeling (ECM) as shown in Fig 1 (b). Leveraging the 3D physical world as a bridge, ECM builds explicit pixel-wise correspondence across multi-view and multi-frame. This approach decouples the model from overfitting to specific camera parameters for multi-view and breaks the chronological order for multi-frame, thus unleashing the generalization capability across spatial-level (Fig 1 (c)) and temporal-level (Fig.1 (d)), even trained on dataset lacking such diversity. Building on ECM's strengths, we further introduce an overlap-based view matching strategy to dynamically select the most relevant context, and a random frame sampling strategy to mitigate the issue of over-reliance on temporal adjacent frames during generation.

For controllable generation, we identify the issue of information loss inherent in existing conditional encoding and injection pipelines, as shown in Fig 4 (a) and (b), and propose an information-preserving control mechanism to alleviate this issue. Furthermore, our control mechanism can be extended to be identity-aware with foreground appearance features from reference frames, yielding better controllability and foreground temporal consistency.

To summary, our contributions are summarized as follows:

- We propose DriveCamSim, a novel generalizable camera simulation framework with the core idea of Explicit Camera Modeling, along with an overlap-based view matching and a random frame sampling strategy. These designs not only enhance visual quality, but also unleash the generalization capability across spatial-level and temporal-level, supporting flexible camera simulation for downstream application.

- We diagnose and address critical information loss in existing conditional pipelines, proposing an information-preserving control mechanism for better controllability, which can be extended to be identity-aware to enhance foreground temporal consistency.

- Through extensive experiments, we demonstrate state-of-the-art performance in visual quality, controllability and generalization capability, with ablation studies validating the efficacy of our key designs.

## 2 RELATED WORKS

### 2.1 CROSS-VIEW INTERACTION FOR MULTI-VIEW IMAGE GENERATION

Effective cross-view interaction is crucial for maintaining spatial consistency in overlapping regions between adjacent camera views. Existing approaches predominantly employ multi-head attention for cross-view modeling, where image patches from one view serve as queries while patches from neighboring views provide keys and values (Yang et al., 2023; Wen et al., 2024b). Recent advancements include MagicDrive (Gao et al., 2023), which incorporates camera parameters as scene-level conditioning, and DriveDreamer-2 (Zhao et al., 2025b), which reformulates cross-view interaction as intra-view processing by concatenating multi-view images along the width dimension. However, these methods inherently assume fixed camera configurations during training, leading to model specialization on specific viewpoint geometries. This fundamental limitation results in constrained generation capability that cannot extrapolate beyond the trained camera parameter distribution, significantly restricting practical deployment scenarios. In contrast, our framework overcomes this limitation by enabling generalization across diverse camera configurations during inference, thereby supporting flexible camera simulation for real-world applications.

### 2.2 CROSS-FRAME INTERACTION FOR MULTI-VIEW VIDEO GENERATION

Maintaining temporal consistency in video generation requires effective cross-frame interaction. While most existing methods employ multi-head attention to model temporal relationships, they rely on spatially aligned patches in 2D image space, which often fail to maintain alignment in the 3D physical world —particularly in high-speed scenarios. This implicit modeling make the model overfit to the specific video frequency in training dataset, limiting their applicability in real-world settings. For instance, DreamForge (Mei et al., 2024) generates 7-frame clips at 12Hz but only utilizes the last frame as input for 2Hz driving agent (Yang et al., 2024), resulting in inefficient computation. Furthermore, while high-frequency training data can be downsampled to produce low-frequency outputs, the reverse is not feasible. In contrast, our approach is able to generalizing across varying frame rates , enabling high-frequency generation from low-frequency training data, and even support generation in reverse temporal order.

### 2.3 CONTROL MECHANISM FOR 3D CONDITION

The control mechanism operates through two sequential stages: (1) the condition encoding stage transforms low-dimensional control signals into high-dimensional condition embeddings, and (2) the condition injection stage incorporates these embeddings into the image latent space. Current approaches can be categorized into two predominant paradigms: **Perspective-based Control** (Wang et al., 2024b; Wen et al., 2024b): As shown in Fig 4 (a), this method projects 3D bounding boxes and road layouts onto 2D perspective views during encoding, followed by direct addition to image latents. However, the 3D-to-2D projection inherently suffers from depth information loss. For instance, a large vehicle at a far distance and a small vehicle at close range may produce similarly sized 2D bounding boxes, introducing ambiguity for model learning. **Attention-based Control** (Gao et al., 2023): As shown in Fig 4 (b), this approach encodes bounding boxes as instance-level embeddings and integrates them via cross-attention mechanisms. While effective in some scenarios,

this paradigm learns implicit view transformations that tend to overfit to specific camera parameters, consequently losing critical relative pose information between objects and the camera. In contrast, our proposed control mechanism systematically preserves spatial and geometric information throughout both encoding and injection stages.

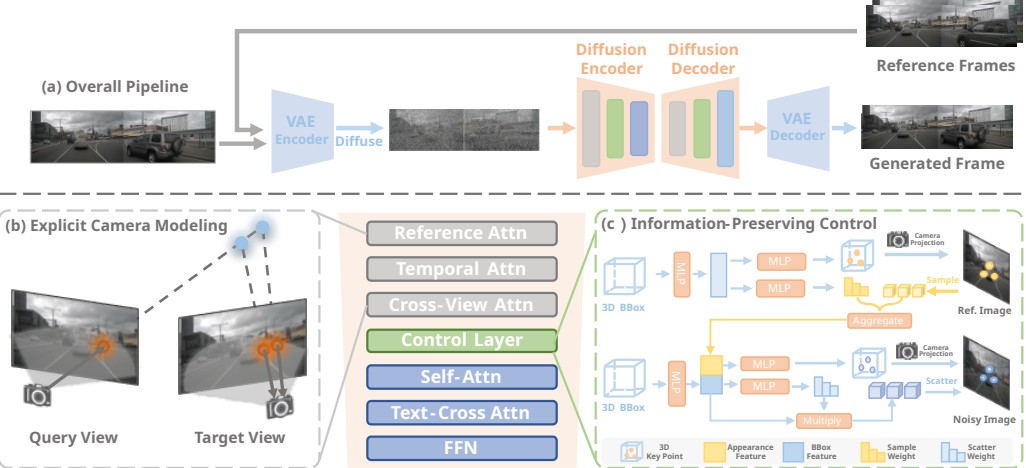

Figure 2: Overall framework of DriveCamSim. The proposed method (a) is built upon a pretrained latent diffusion model with several new layers inserted. Explicit Camera Modeling (b) is proposed for cross-view and cross-frame interaction. Information-Preserving Control (c) is designed to enhance controllability.

## 3 METHODS

### 3.1 PROBLEM FORMULATION

This work addresses the problem of controllable camera simulation for autonomous driving. Following Bench2Drive-R (You et al., 2024), at a given time $t$, below information is provided as the input of our generative model:

1. **3D bounding boxes**: $\mathbf{B}_t = \{(b_i, c_i)\}_{i=1}^{N_b}$, where $b_i = (x_i, y_i, z_i, l_i, w_i, h_i, yaw_i)$ is the bounding boxes for foreground objects (vehicles, pedestrians, bicycles, etc.) within a specific range; $c_i \in \mathcal{C}_{box}$ is the semantic label.

2. **Vectorized map elements**: $\mathbf{M}_t = \{(v_i, c_i)\}_{i=1}^{N_m}$, where $v_i = (x_j, y_j)_{j=1}^{N_v}$ represents vertices for polygon map elements (cross-walk regions, etc.) and interior points for linestring map elements (road boundaries, lane dividers, etc.); $c_i \in \mathcal{C}_{map}$ represents the map class.

3. **Ego pose**: $\mathbf{E}_t \in \mathbb{R}^{4 \times 4}$ is ego pose matrix including ego-to-global translation and rotation.

4. **Camera parameters**: $\mathbf{K} = \{\mathbf{K}_i \in \mathbb{R}^{4 \times 4}\}_{i=1}^{N_{cam}}$, where $\mathbf{K}_i$ is the camera transformation matrix composed of intrinsic and extrinsic matrices that transforms points from ego coordinate system to image coordinate system.

5. **Reference information**: $\mathbf{H}_r = \{(\mathbf{I}_r, \mathbf{K}_r, \mathbf{E}_r, \mathbf{B}_r)\}_{r=1}^{N_r}$, where $N_r$ is the number of reference frames. The reference information includes original recorded images, camera parameters, corresponding pose and boxes, which are used to retrieve box appearance feature.

6. **Historical information**: $\mathbf{H}_h = \{(\mathbf{I}_h, \mathbf{K}_h, \mathbf{E}_h, \mathbf{B}_h)\}_{h=1}^{N_h}$, where $N_h$ is the number of historical frames. $\mathbf{H}_h$ is similar to $\mathbf{H}_r$, except that the reference images $I_r$ are logged real images, while historical images $I_h$ are previous generated images.

With these information, out model generate multi-view images at time $t$: $I_t = \mathcal{G}(\mathbf{B}_t, \mathbf{M}_t, \mathbf{E}_t, \mathbf{K}, \mathbf{H}_r, \mathbf{H}_h)$, which will be used as historical information for auto-regressive gen-

eration. We adopt such an online generation scheme rather than offline long video generation to enable reactive simulation for downstream AD algorithms.

## 3.2 OVERALL FRAMEWORK

The overall framework of DriveCamSim is shown in Fig 2. Our model builds upon a pretrained latent diffusion model, with several attention layers and control layers inserted within attention blocks.

## 3.3 EXPLICIT CAMERA MODELING

The motivation for Explicit Camera Modeling is to build correspondence between pixels across multi-view and multi-frame, enabling interaction in 3D physical world rather than 2D image space. For simplicity, we take a query view $V_{query}$ and a target view $V_{key}$ to illustrate ECM, but can easily be extended to multi views. Query view is selected from current frame, while target view can be selected from current frame (for cross-view attention), reference frame (for reference attention) or historical frame (for temporal attention).

**Building Pixel Correspondence.** For each pixel $p_q = (u_q, v_q)$ in $V_{query}$, we first project it to 3D space. However, regressing to a precise depth value is difficult, especially for noisy latents. So we set several depth anchors $d = \{d_i\}_{i=1}^{D}$ and back project $p_q$ to 3D points $\{P_{qi}\}_{i=1}^{D}$, where $P_{qi} = d_i \cdot K^{-1} \cdot p_q$. $\{P_{qi}\}$ are further projected to $V_{key}$ to get $\{p_{ki}\}_{i=1}^{D}$, where $p_{ki} = K_k \cdot E_k^{-1} \cdot E_t \cdot P_{qi}$, $K_k$ and $E_k$ are camera projection matrix and global pose of target view. By doing so, we build correspondence between query view pixel $p_q$ and target view pixels $\{p_{ki}\}$.

**Feature Aggregation.** After building pixel correspondence, we aggregate features at $\{p_{ki}\}$ to refine query feature at $p_q$. For each $p_q$, we have $d$ target pixels, considering not all target pixels are equally important, we predict a depth distribution to model the attention weights between $p_q$ and $\{p_{ki}\}$ with $W_{qk} = \text{Softmax}(\text{MLP}(f_q)) \in \mathbb{R}^d$, where $f_q = x_q(u, v)$ is the query pixel feature and $x_q$ is the feature of query view. We also note that 3D points $\{P_{qi}\}$ may project outside of $V_{key}$, so we filter out these outlier points by setting corresponding weights to zero. Then we conduct image interaction by updating query feature with $f_q = f_q + \sum_{i=1}^{D}(W_{qki} \times f_{ki})$, where $f_{ki}$ is target pixel feature at $p_{ki}$.

**Overlap-based Target View Matching.** Now for query view $V_{query} = V_{n,t}$ where $n \in \{1, ..., N_{cam}\}$ is index of view, we extend the target view number to more than one. One problem raises that: how to choose the target view? One naive strategy is to choose $\{V_{n-1,t}, V_{n+1,t}\}$ for cross-view attention, $\{V_{n,r}\}$ for reference attention, and $\{V_{n,h}\}$ for temporal attention. However, in scenarios like turning at intersection, $\{V_{n,r}\}$ and $\{V_{n,h}\}$ might have a small overlap with $V_{n,t}$, resulting in invalid computation. To address this, we propose an overlap-based target view matching strategy to dynamically search best target views. We notice that the ineffective computation comes

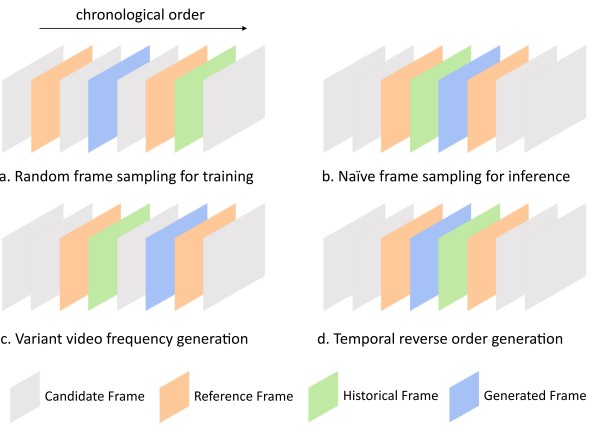

Figure 3: Frame sampling strategy for training and inference.

from much zero weights for outlier points, so we use the percentage of $\{p_{ki}\}$ that hit on target view to represent the degree of overlap, and select views that have maximum overlap with query view as target views. This strategy benefit the feature interaction by providing most relevant context from target views.

**Frame Sampling.** Another problem arise that how to sample reference frame and historical frame. One naive method is to sample frames following chronological order. However, we found these adjacent frames share similar context, resulting in over-reliance on adjacent frames when generating current frame. As shown in Fig 3 (a), built upon explicit camera modeling, our model breaks chronological order in multi-frame video, enabling a random sampling strategy at training to force the model learn the geometric transformation from historical and reference frames to generation frame, rather than simply copy the pattern. This training strategy also unleash flexible inference schemes in Fig 3 (b-d), e.g. generation with variant video frequency or temporal reverse order.

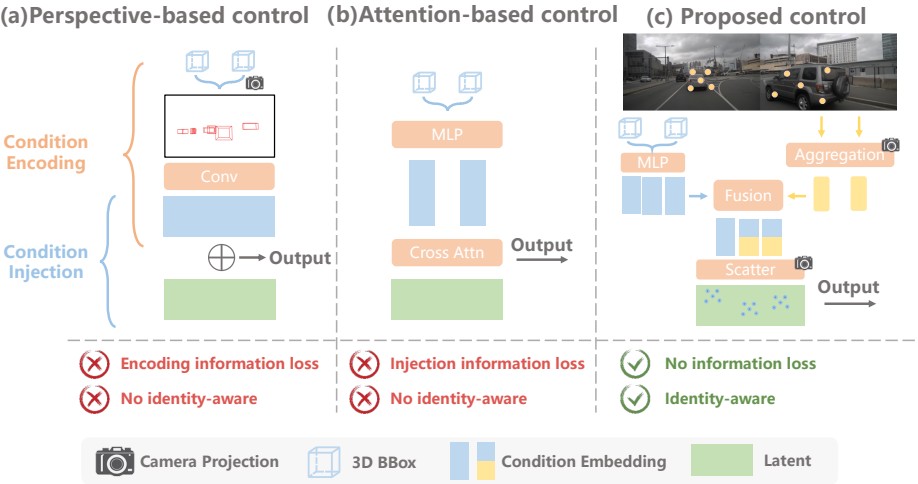

Figure 4: Compared with perspective-based(Yang et al., 2023) and attention-based control(Gao et al., 2023), our control mechanism preserves information in encoding and injection stage, and support identity feature encoding.

## 3.4 CONDITIONAL MECHANISM

**Text Condition.** Following common practice (Gao et al., 2023), our model uses text description for scene-level control. We build a simple prompt template as "A driving scene image. {Weather}. {Daytime}." The prompt is embedded with CLIP text encoder and injected into image latent through text cross attention.

**3D Bonding Boxes Encoding.** To prevent information loss in 3D-to-2D projection, we directly encode boxes and class label into an instance-level embedding following MagicDrive (Gao et al., 2023):

$$E_{box_i} = \mathrm{MLP}(b_i) + \mathrm{CLIP}(c_i)$$

**Extension to be Identity-Aware.** Previous methods only encode geometric information from $b_i$ and semantic information from $c_i$, lacking identity information and let the model learn to match foreground objects from different frames. This may be confusing in some cases, e.g. crowded scenes. To be completely controllable, we additionally encode appearance feature from historical and reference frames. Following sparse-centric perception model (Lin et al., 2022), we similarly encode the box $b_h$ with same identity at historical frame (or reference frame ), then use the box embedding $E_{box_h}$ to generate several keypoints $\{P_j\}_{j=1}^{N_j}$ around the box and corresponding attention weights $\{w_j\}_{j=1}^{N_j}$, then project $\{P_j\}$ to historical frame with $p_j = K_j \cdot P_j$ to sample the feature $\{f_{p_j}\}$, and aggregate appearance feature as $A_{box_h} = \sum_{j=1}^{N_j} w_j \cdot f_{p_j}$. The box embedding is then

Table 1: Comparisons of realism and controllability on nuScenes validation set. * means using real images as reference.

| Method | FID | BEVFusion(Camera Branch) | | | | StreamPETRstreampetr | |
| | | NDS↑ | mAP↑ | mAOE↓ | mIoU↑ | NDS↑ | mAP↑ |
|---|---|---|---|---|---|---|---|
| Oracle | - | 41.20 | 35.53 | 0.56 | 57.09 | 57.10 | 48.20 |
| BEVControl | 24.85 | - | - | - | - | - | - |
| MagicDrive | 16.20 | 23.35 | 12.54 | 0.77 | 28.94 | 35.51 | 21.41 |
| Panacea | 16.69 | - | - | - | - | 32.10 | - |
| Panacea+ | 15.50 | - | - | - | - | 34.60 | - |
| DriveCamSim | **14.07** | **23.87** | **12.75** | **0.64** | **34.84** | **39.49** | **22.41** |
| Bench2Drive-R* | 10.95 | 25.75 | 13.53 | 0.73 | 42.75 | 40.23 | 24.04 |
| DriveCamSim* | **7.86** | **26.55** | **14.47** | **0.67** | **43.36** | **44.16** | **28.16** |

Table 2: Performance of UniAD's different tasks on nuScenes validation set. * means using real images as reference.

| Method | Detection | | BEV Segmentation | | | | Planning | | Occupancy |
| | NDS↑ | mAP↑ | Lanes↑ | Drivable↑ | Divider↑ | Crossing↑ | avg.L2(m)↓ | avg.Col.(%)↓ | mIoU↑ |
|---|---|---|---|---|---|---|---|---|---|
| Oracle | 49.85 | 37.98 | 31.31 | 69.14 | 25.93 | 14.36 | 1.05 | 0.29 | 63.7 |
| MagicDrive | 29.35 | 14.09 | 23.73 | 55.28 | 18.83 | 6.57 | 1.18 | **0.33** | 54.6 |
| DriveCamSim | **31.55** | **14.70** | **25.86** | **56.44** | **20.66** | **8.50** | **1.16** | 0.40 | **55.7** |
| Bench2Drive-R* | 33.04 | 15.16 | 25.50 | 56.53 | **21.27** | 8.67 | **1.15** | **0.31** | 55.5 |
| DriveCamSim* | **34.88** | **16.90** | **26.31** | **58.58** | 21.25 | **9.16** | **1.15** | 0.40 | **57.0** |

updated as:

$$E_{box_i} = \mathrm{MLP}(\mathrm{b_i}) + \mathrm{CLIP}(c_i) + \sum_{h=1}^{N_h} A_{box_h} + \sum_{r=1}^{N_r} A_{box_r}$$

**Scatter-based Condition Injection.** To be compatible to our ECM and generalize across different camera parameters, we need to project the condition embedding onto image latent using camera parameters. However, the instance-level embedding is not suitable to directly add to image latents. To address this, we propose a scatter-based condition injection method, which can be regarded as an inverse operation of aggregation. Specifically, we use the condition embedding $E_{box_i}$ to predict several keypoints $\{P_m\}_{m=1}^{N_m}$ around the 3D box $b_i$, and corresponding weights $\{w_m\}_{m=1}^{N_m}$ for each point, the keypoints are projected to image with $p_m = (u_m, v_m) = K_t \cdot P_m$ to find the location on image latent, then the condition embedding is scaled by weights and scatter back to image latent $x$ with $x(u_m, v_m) = x(u_m, v_m) + w_m \cdot E_{box_i}$. In practice, $(u_m, v_m)$ are not integers, so we use bilinear scatter similar to bilinear sampling.

### 3.4.1 VECTORIZED MAP ELEMENTS.

Vectorized map elements are encoded similarly to boxes, and our scatter-based method also applies to map condition injection.

$$E_{vec_i} = \mathrm{MLP}(m_i) + \mathrm{CLIP}(c_i)$$

## 4 EXPERIMENTS

### 4.1 EXPERIMENTAL SETUPS

**Dataset and Baselines.** We employ nuScenes dataset (Caesar et al., 2020), which have 700 street-view scenes for training and 150 for validation with 2Hz annotation. Our baseline models include image generation methods (BEVControl, (Yang et al., 2023), MagicDrive (Gao et al., 2023)), video generation methods (Panacea (Wen et al., 2024b), Panecea+ (Wen et al., 2024a)) and simulation-oriented method with real images as reference (Bench2Drive-R (You et al., 2024)).

**Evaluation Metrics.** We evaluate the generation realism with Frechet Inception Distance (FID). For controllability, we use BEVFusion (Liu et al., 2023) to evaluate foreground object detection and

Table 3: Performance of SparseDrive on two generated datasets with original and perturbed camera parameters. Smaller performance gap indicates generative model's robust generalization ability on camera parameter variations. "-P" means the metric on perturbed dataset.

| ID | Control Mechanism | 3DOD NDS/NDS-P↑ | 3DOD mAP/mAP-P↑ | Tracking AMOTA/AMOTA-P↑ | Online Mapping mAP/mAP-P↑ | Planning L2/L2-P↓ | Planning Col/Col-P(%)↓ |
|---|---|---|---|---|---|---|---|
| 1 | Ours | **36.44/35.80** | **19.57/19.08** | **9.05/9.47** | **22.84/22.02** | **0.69/0.98** | **0.19/0.42** |
| 2 | Perspective-based (Yang et al., 2023) | 31.15/30.11 | 14.26/13.54 | 5.52/5.30 | 21.36/18.18 | 0.73/1.04 | 0.20/0.46 |
| 3 | Attention-based (Gao et al., 2023) | 26.23/25.04 | 10.01/8.96 | 4.92/3.74 | 15.65/13.92 | 0.81/1.05 | 0.32/0.43 |

Table 4: Generalization capability across different datasets.

| ID | Training dataset | FID | 3DOD: NDS | Online Mapping: mAP |
|---|---|---|---|---|
| 1 | nuPlan | 31.95 | 16.26 | 2.99 |
| 2 | nuScenes | 15.70 | 36.44 | 22.84 |
| 3 | nuPlan + nuScenes | **15.50** | **38.23** | **24.26** |

background map segmentation, StreamPETR (Wang et al., 2023b) to evaluate temporal consistency of generated image sequences, and UniAD (Hu et al., 2023) for end-to-end planning.

**Model Setup.** We utilizes pretrained weights from Stable Diffusion v1.5 (Rombach et al., 2022), as we do not have trainable copy from ControlNet (Zhang et al., 2023), we train all parameters of UNet (Ronneberger et al., 2015). The generation resolution is 224×400, and images are sampled using UniPC (Zhao et al., 2023) scheduler for 20 steps with CFG at 2.0. Through we made a distinction between reference frames and historical frames, our explicit camera modeling can handle them in a unified format, enabling flexible inference mode. We set total frames up to 3 ($N_r + N_h = 3$), and use 3 historical frames as input by default. For comparison with Bench2Drive-R (You et al., 2024), we use 1 historical frame and 2 reference frames within recordings with closest distance.

## 4.2 MAIN RESULTS

**Generation Realism and Controllability.** As show in Tab 1, our method outperforms baselines in generation realism with a lower FID score, and achieves better controllability on both foreground and background generation.

**Temporal Consistency.** As show in Tab 1, perception results evaluted by StreamPETR (Wang et al., 2023b) are notably better than the baseline methods, whether with or without reference images. This demonstrates the temporal consistency of our auto-regressive generated image sequences.

**Generation for End-to-End Planning.** As show in Tab 2, our method outperforms baselines on nearly all metrics, indicating the potential of our method for driving agent simulation.

## 4.3 GENERALIZATION CAPABILITY

**Camera Parameter Generalization.**

To evaluate generalization capabilities, we employ a SOTA end-to-end driving model, SparseDrive (Sun et al., 2024), across multiple tasks. SparseDrive is trained using data augmentation techniques—including random resizing, cropping, and rotation—which inherently enhance its robustness to minor perturbations in camera parameters. Accordingly, we use our generative model to produce two distinct datasets: one with the original camera parameters of nuScenes and another with randomly perturbed parameters. Since the perturbation strategy aligns with the augmentation techniques used in SparseDrive's training, we can fairly compare performance between the two datasets to assess the generative model's generalization ability. As shown in Table 3, our method not only achieves the best performance under original camera settings but also maintains strong controllability under perturbed conditions, demonstrating robust generalization across varying camera parameters.

**Generalization across Datasets.** We further add experiments on nuPlan dataset (Caesar et al., 2021) with different camera rigs to show the generalization capability. We train our model on different

Table 5: Ablation for explicit and implicit camera modeling. ECM-S, ECM-T and ECM-R represent explicit camera modeling for cross-view, cross-frame and reference attention. The implicit camera modeling follows Panacea.

| ID | ECM-S | ECM-T & ECM-R | 3DOD | | Tracking | Online Mapping | Planning | |
|----|-------|---------------|------|------|----------|----------------|----------|------|
| | | | NDS↑ | mAP↑ | AMOTA↑ | mAP↑ | L2↓ | Col(%)↓ |
| 1 | ✓ | ✓ | **36.44** | **19.57** | **9.05** | **22.84** | **0.69** | **0.19** |
| 2 | | ✓ | 33.83 | 16.01 | 6.76 | 20.26 | 0.79 | 0.28 |
| 3 | ✓ | | 30.67 | 14.84 | 6.08 | 12.95 | 0.83 | 0.36 |

Table 6: Ablation for overlap-based view matching (OVM) and random frame sampling strategy.

| ID | OVM at training/infernce | Random Frame Sampling | 3DOD | | Tracking | Online Mapping | Planning | |
|----|--------------------------|------------------------|------|------|----------|----------------|----------|------|
| | | | NDS↑ | mAP↑ | AMOTA↑ | mAP↑ | L2↓ | Col(%)↓ |
| 1 | ✓ / ✓ | ✓ | **36.44** | **19.57** | **9.05** | **22.84** | **0.69** | 0.19 |
| 2 | ✓ / ✗ | ✓ | 36.25 | 19.40 | 8.91 | 22.29 | **0.69** | **0.18** |
| 3 | ✗ / ✗ | ✓ | 33.58 | 16.11 | 6.66 | 18.74 | 0.79 | 0.29 |
| 4 | ✓ / ✓ | | 32.90 | 15.24 | 7.00 | 16.95 | 0.72 | 0.28 |

Table 7: Ablation for control mechanism and identity feature. Perspective-based control is from Yang et al. (2023) and attention-based control is from Gao et al. (2023)

| ID | Control Mechansim | Identity Aware | 3DOD | | Tracking | Online Mapping | Planning | |
|----|-------------------|----------------|------|------|----------|----------------|----------|------|
| | | | NDS↑ | mAP↑ | AMOTA↑ | mAP↑ | L2↓ | Col(%)↓ |
| 1 | Our Control | ✓ | **36.44** | **19.57** | **9.05** | **22.84** | **0.69** | **0.19** |
| 2 | Our Control | | 34.96 | 17.16 | 7.22 | 20.24 | 0.78 | 0.32 |
| 3 | Perspective-based control | | 31.15 | 14.26 | 5.52 | 21.36 | 0.73 | 0.20 |
| 4 | Attention-based control | | 26.23 | 10.01 | 4.92 | 15.65 | 0.81 | 0.32 |

datasets and evaluate the model on nuScenes. As shown in the Table 4, with our explicit camera modeling and control mechanism, even trained only on nuPlan (8 views), our model could achieve a certain degree of foreground controllability on nuScenes (6 views). Since the object classes are not exactly the same for nuScenes and nuPlan, the NDS for detection lags behind the model directly trained on nuScenes. Due to the two datasets collect data from different cities, the background controllability (online mapping mAP) is not good. However, when combine these two datasets for joint training, even the data distribution, scene style and the camera rigs of two datasets are different, the model achieves better performance compared with the model trained only on nuScenes.

### 4.4 ABLATION STUDIES

**Ablation for Camera Modeling.** As demonstrated in Table 5, replacing our explicit camera modeling with implicit camera modeling leads to consistent performance degradation across all evaluation metrics, especially for temporal and reference attention, indicating the importance of aligning in 3D physical world rather than 2D image space.

**Ablation for Overlap-based View Matching and Random Frame Sampling Strategy.** As illustrated in Table 6, the ablation study reveals key observations as follows. When overlap-based view matching (OVM) is utilized during training but disabled at inference (ID-2), a marginal performance degradation occurs in all perception tasks. And complete removal of OVM during both training and inference (ID-3) leads to a more pronounced performance drop, underscoring its importance. The exclusion of random frame sampling during training (ID-4) further adversely affects task performance, suggesting its importance for model learning.

**Ablation for Control Mechanism.** As show in Tab 7, compared to ID-2, ID-1 introduces appearance feature and brings improvement on tracking metric, indicating better foreground temporal consistency. ID-3 indicates that it's necessary to preserve 3D information in condition encoding, and ID-4 shows attention-based control suffers from slow convergence for losing view transformation information between boxes and cameras.

## 4.5 QUALITATIVE RESULTS

We compare our method with MagicDrive (Gao et al., 2023) and DreamForge (Mei et al., 2024) for spatial-level generalization capability in Fig 5. Taking the example of rotating the front camera 20° to the left, we can find that with implicit camera modeling and attention-based control, MagicDrive generates nearly same images before and after rotation. DreamForge, enhanced with perspective-based control, maintains foreground controllability after rotation, but fails to generate correct background. Our method, with explicit camera modeling and information-preserving control, correctly handles both foreground and background. Additional visualizations illustrating spatial and temporal generalization are provided in the Appendix E.

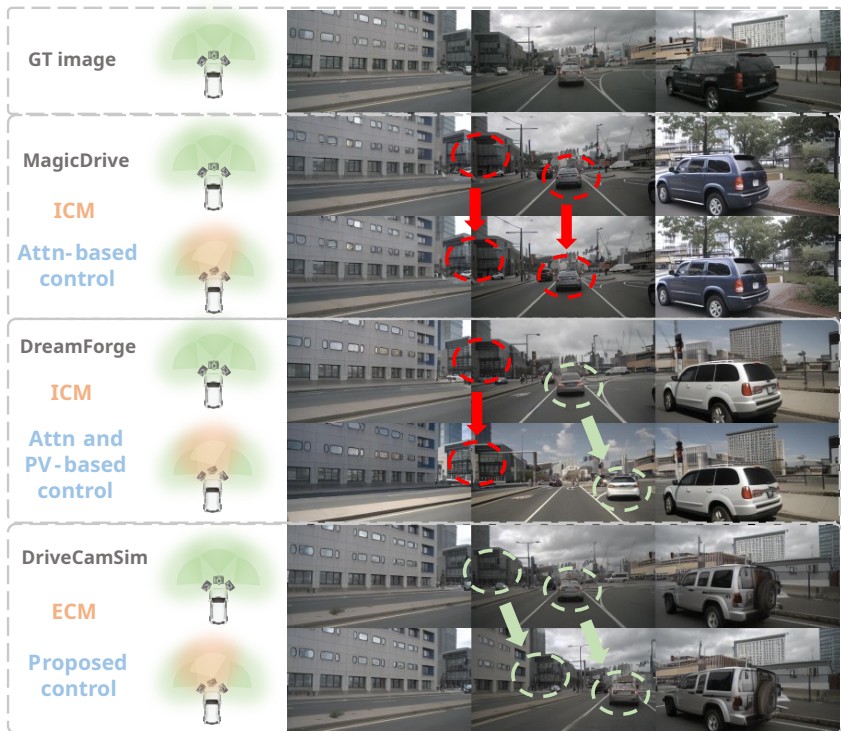

Figure 5: Qualitative results for spatial-level generalization. Rotate front camera 20° to the left, DriveCamSim succeed to generate images with correct foreground and background, while Magic-Drive and DreamForge fails.

## 5 CONCLUSION

In this work, we explore the explicit camera modeling and information-preserving control mechanism for controllable camera simulation in driving scene. The resulting framework DriveCamSim achieves SOTA visual quality and controllability, while unleashing the spatial and temporal-level generalization capability, enabling flexiable camera simulation for downstream application. We hope that DriveCamSim can inspire the community to rethink physically-grounded camera modeling paradigms for driving simulation.

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

## A   MORE RELATED WORKS

There is another line of research trying to solve the novel view synthesis problem with generative models, exemplified by FreeVS (Wang et al., 2024a), FreeSim (Fan et al., 2025), DriveDreamer4D (Zhao et al., 2025a) and StreetCrafter (Yan et al., 2025). Below we clarify the conceptual and methodological gap between these approaches and ours.

- Problem Scope. These methods address novel ego-pose synthesis in a single-view regime; multi-view imagery is obtained by repeatedly forwarding the same model. In contrast, we target simultaneous multi-view generation under arbitrary camera rigs, which imposes an additional cross-view consistency constraint that must be satisfied in a single forward pass.

- Methodological Trade-offs. Our goal is to build a scalable, reactive simulator with generative model. DriveDreamer4D first generates a video clip and then reconstructs the scene with 4D Gaussian Splatting while FreeSim proposes a progressive reconstruction strategy to combine the generation and reconstruction model. The rendering stage yields excellent temporal coherence, yet the optimization is performed offline and not pure generative. StreetCrafter and FreeVS lift colored point clouds into the target view via depth-guided warping. The availability of metric depth substantially simplifies the geometric reasoning task, but restricts the methods to logged sequences that contain synchronized LiDAR. Our pipeline consumes only images, enabling training on massive, low-cost image collections and permitting seamless switching between "imagining" a new scene or "recovering" an existing one.

## B   EXPERIMENTAL REPRODUCIBILITY DETAILS

We train all parameters on 16 RTX 4090 GPUs using AdamW (Loshchilov & Hutter, 2017) optimizer with a linear warm-up of 3000 iterations and learning rate of 2e-4, the total batch size is $16 \times 4 = 64$. The model is trained for 400 epochs in main results and 100 epochs in ablation studies. We only use 2Hz data in nuScenes for training. For building pixel correspondence, we set 10 fixed depth anchor in range of [1, 60] with linear increasing discretization (Tang et al., 2020). For overlap-based target view matching, we set the number of target views to twice the number of frames, which is 2 for cross-view attention, and $2 \times (N_r + N_h)$ for reference and temporal attention. For random frame sampling, we randomly sample 4 frames (3 for reference and historical frames and 1 for generation frame) within 12 consecutive frames.

## C   MORE EXPERIMENTS

### C.1   TEMPORAL CONSISTENCY

To evaluate the temporal consistency of generated videos, we provide results on W-CODA(Chen et al., 2025) benchmark in Table 8. Our method surpasses DreamForge (Mei et al., 2024), which ranks second on the benchmark, while the first place solution DiVE (Jiang et al., 2024b) adopts a more advanced DiT-based pretrained model.

### C.2   TRAINING SUPPORT

We use SparseDrive (Sun et al., 2024) stage-1 as a perception model to evaluate the training support of generated videos. Trained solely on generated videos, SparseDrive achieves 41.03 NDS and 48.14 mAP, only slightly lags behind training on real data. Using synthetic videos for data augmentation can further boost performance to surpass the model trained on real data.

### C.3   CAMERA POSE ESTIMATION

To quantitatively verify how good the generated images follow the input camera parameters, we conduct a camera pose estimation experiment using generated images. We adopt VGGT (Wang et al., 2025) as a modern feed-forward SfM model and follow PoseDiffusion (Wang et al., 2023a) to report Relative Rotation Accuracy (RRA) and Relative Translation Accuracy (RTA) at two thresholds (@15, @30). As shown in Table 10, ID-3, with rotated camera parameters, attains comparable RRA@30 and better RRA@15 than ID-2. We think this is because the rotation enlarges the overlap between neighbor views, which is small for original camera parameters. To validate this, we further conduct ID-4 to generate images with two additional virtual views, resulting in better RRA. In contrast, RTA appears unreliable in this setting—VGGT yields higher RTA on our generated images than on the real images—so we consider RRA as the primary metric.

Table 8: FVD metric on W-CODA benchmark.

| Method | FVD |
|---|---|
| DiVE | 94.5979 |
| DreamForge | 224.7638 |
| DriveCamSim | 195.5768 |

Table 9: Comparison about training support for perception model SparseDrive on detection (NDS) and online mapping (mAP).

| Real | Generated | NDS | mAP |
|---|---|---|---|
| ✓ |  | 45.58 | 51.77 |
|  | ✓ | 41.03 | 48.14 |
| ✓ | ✓ | 49.13 | 53.65 |

## D   MORE DISCUSSION

### D.1   USE OF LARGE LANGUAGE MODELS

The large language model (LLM) was employed as a general purpose writing assistant during the preparation of this manuscript. Its use was limited to language refinement and style adjustment. It was not involved in whole research process.

Table 10: Camera pose estimation results evaluated by VGGT.

| ID | Image Source | Camera Pose | RRA@30↑ | RTA@30↑ | RRA@15↑ | RTA@15↑ |
|---|---|---|---|---|---|---|
| 1 | Real | Original | 0.9089 | 0.4400 | 0.8467 | 0.2049 |
| 2 | Generated | Original | 0.5187 | 0.4858 | 0.2191 | 0.2622 |
| 3 | Generated | Rotation | 0.5133 | 0.4329 | 0.2671 | 0.2240 |
| 4 | Generated | Add virtual view | 0.6010 | 0.3998 | 0.2926 | 0.1957 |

## D.2 LIMITATIONS

Although generalize to camera parameters with small perturbation, we found large perturbation like large translation and rotation in $x$ and $z$ axis result in poor generation result. Despite our effort to combine the advantage of diffusion-based model (benefit from scaling law, no artifacts for complex maneuver) and rendering-based method (flexible novel view synthesis), the view consistency of our method is not as good as rendering-based method. This indicates that there are still potential for improvement in sensor modeling. We leave these problems for future exploration.

## D.3 SOCIAL IMPACT

Our method provide a promising solution for industry, that an foundation generative model can be trained with unifying the data from vehicles that have different camera configurations. And this model can be utilized for data augmentation or evaluation for developing the downstream models that need new viewpoint image as input.

## E MORE VISUALIZATION RESULTS

### E.1 INFORMATION LOSS EXAMPLES

We provide visualization examples to illustrate the information loss issue for perspective-based control. As shown in Fig 6 (a), when we alter the 3D box conditions while keeping the projected 2D boxes identical, the generated foreground always remains the same, suffering from depth information loss. The 3D-to-2D projection process also discards yaw information, resulting in wrong heading for generated vehicles, as shown in Fig 6 (b).

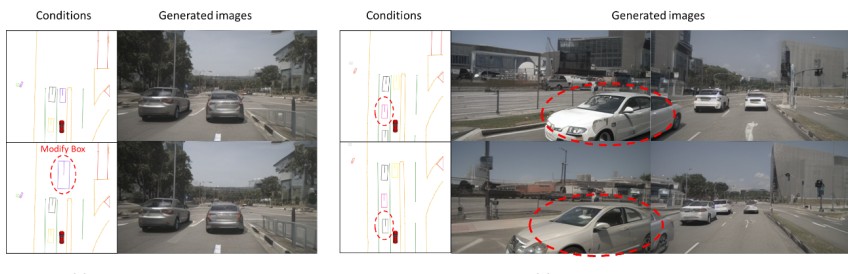

Figure 6: Pespective-based control suffers from (a) depth information loss and (b) yaw information loss in 3D-to-2D projection process.

### E.2 SPATIAL-LEVEL AND TEMPORAL-LEVEL GENERALIZATION

We provide more visualization results for spatial-level and temporal-level generalization.

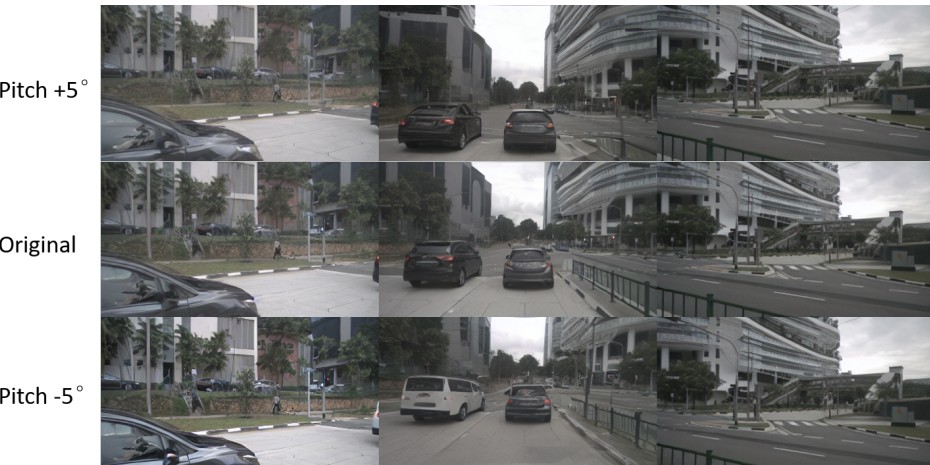

Figure 7: Visualization results for rotating front camera along x-axis.

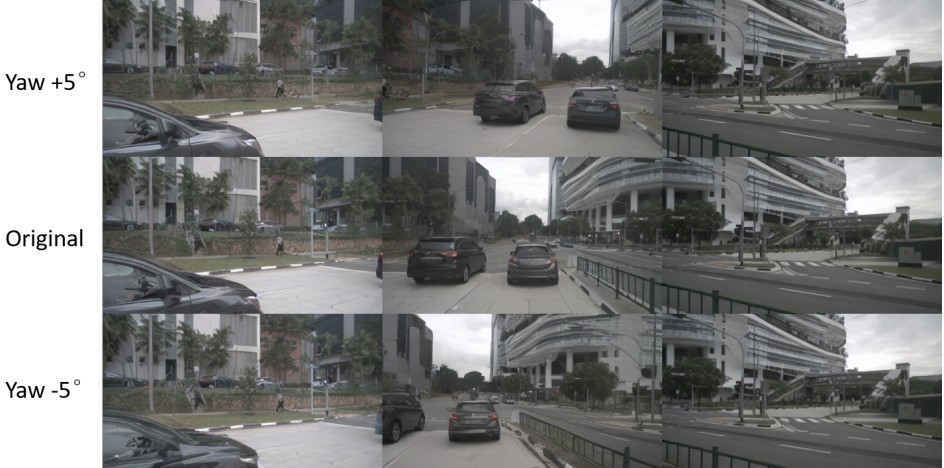

Figure 8: Visualization results for rotating front camera along y-axis.

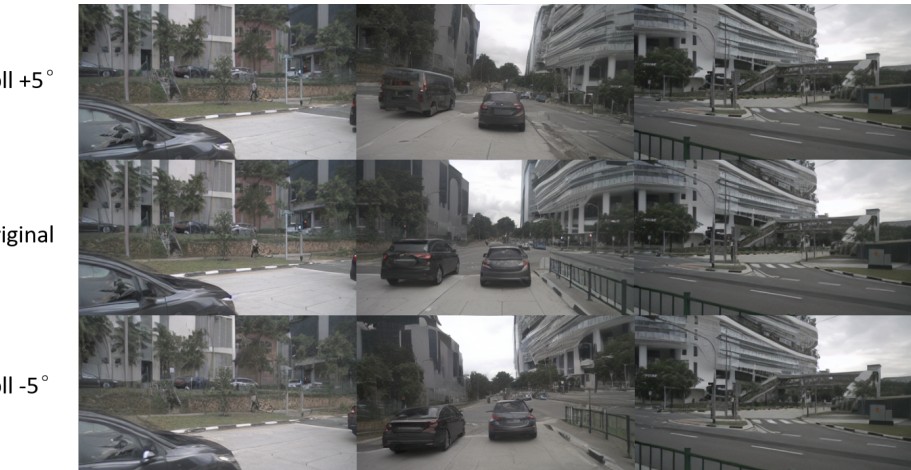

Figure 9: Visualization results for rotating front camera along z-axis.

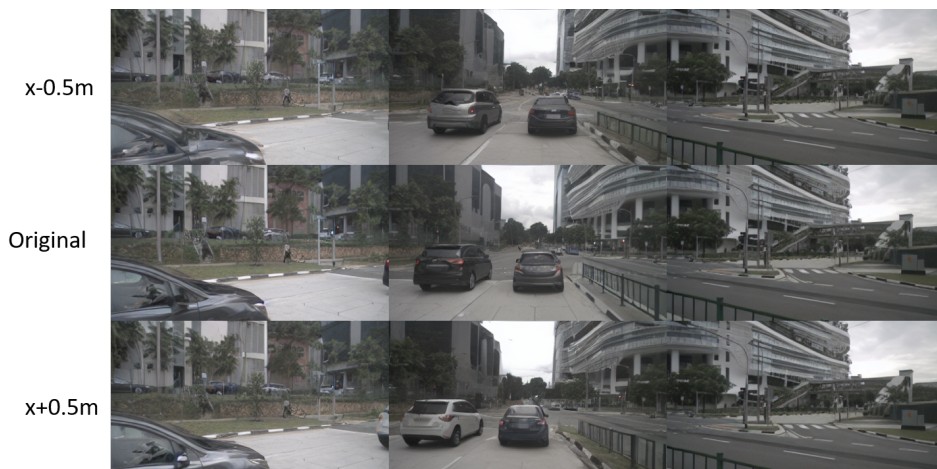

Figure 10: Visualization results for translating front camera along x-axis.

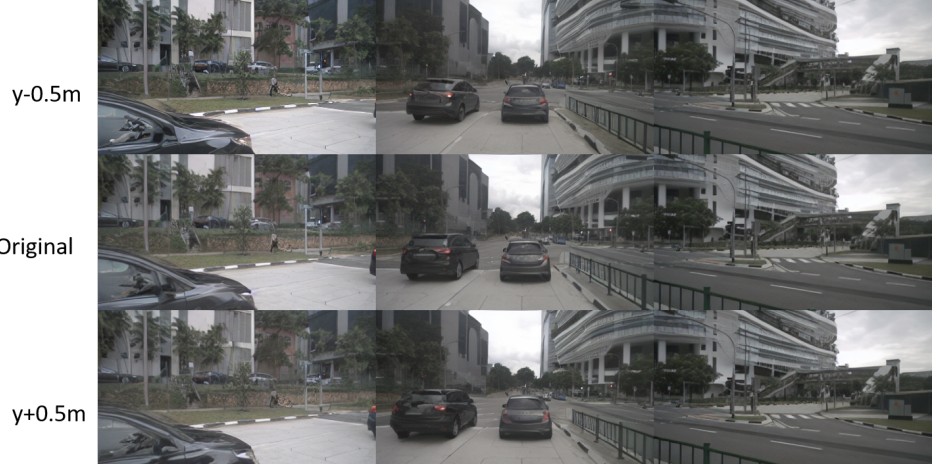

Figure 11: Visualization results for translating front camera along y-axis.

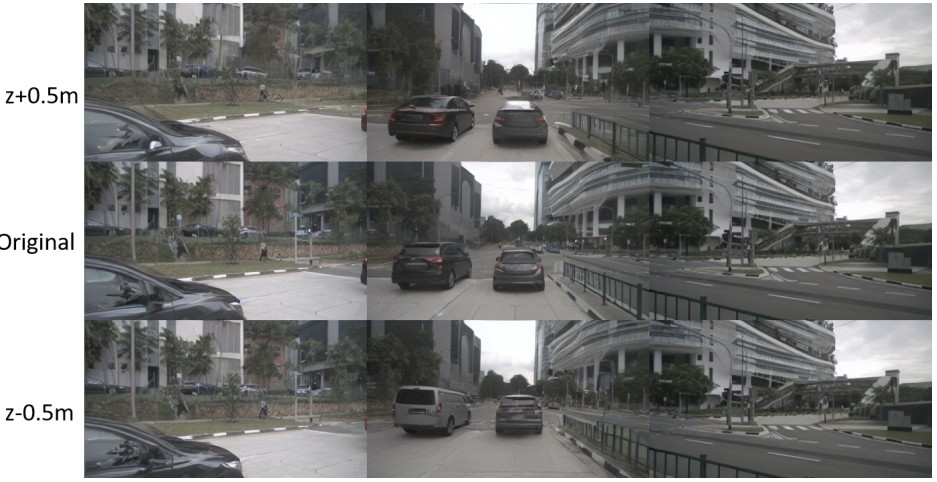

Figure 12: Visualization results for translating front camera along z-axis.

Focal*1.2

Original

Focal*0.8

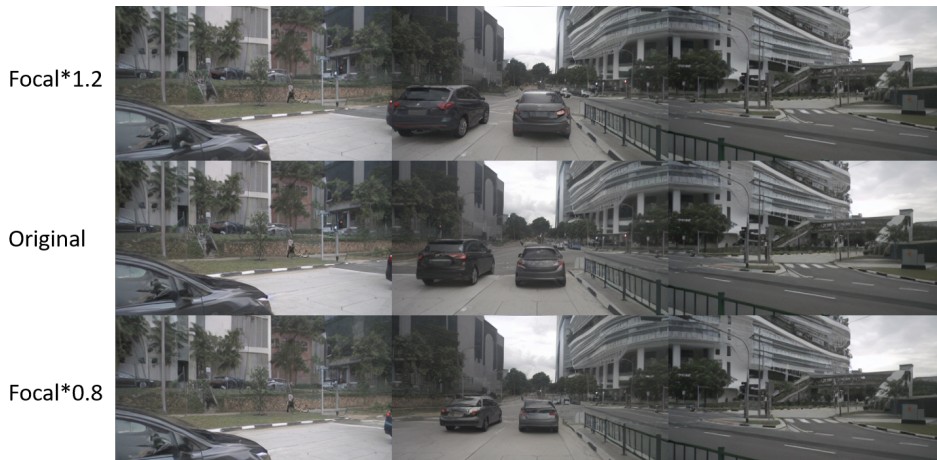

Figure 13: Visualization results for scaling focal length.

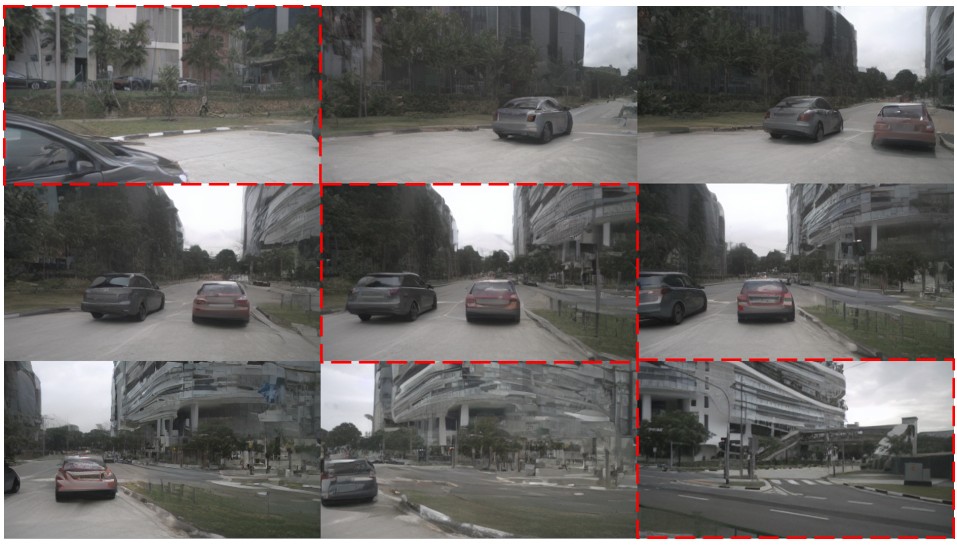

Figure 14: Visualization results for inserting 3 virtual cameras on both sides of the front camera with different yaw angle. 3 views with red border are front-left camera, front camera and front-right camera of nuScenes dataset, while others are virtual cameras.

Original

Pitch +10°

Roll +15°

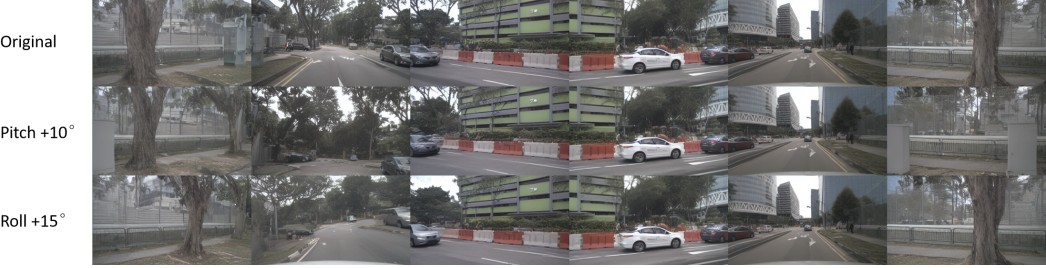

Figure 15: Failure cases of large rotation of front camera.

2Hz

12Hz

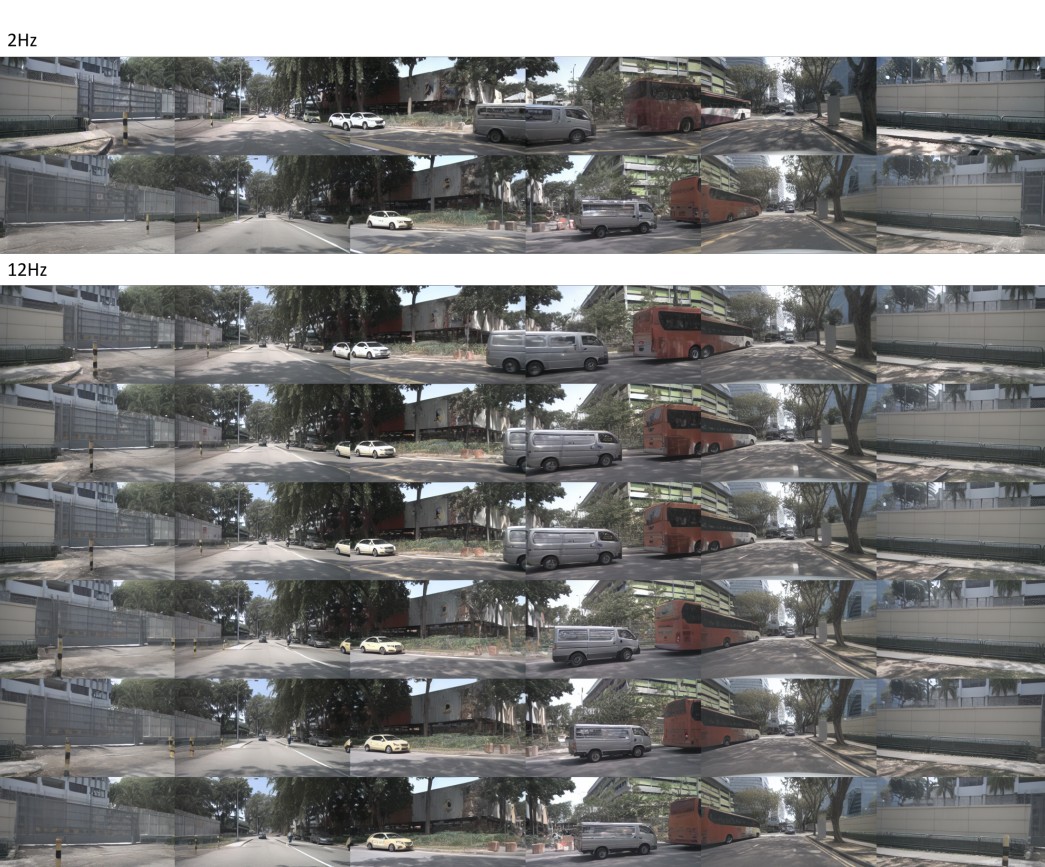

Figure 16: Qualitative results for temporal-level generalization. Trained on 2Hz data, our model can generalize to high-frequency 12Hz generation.

Chronological order

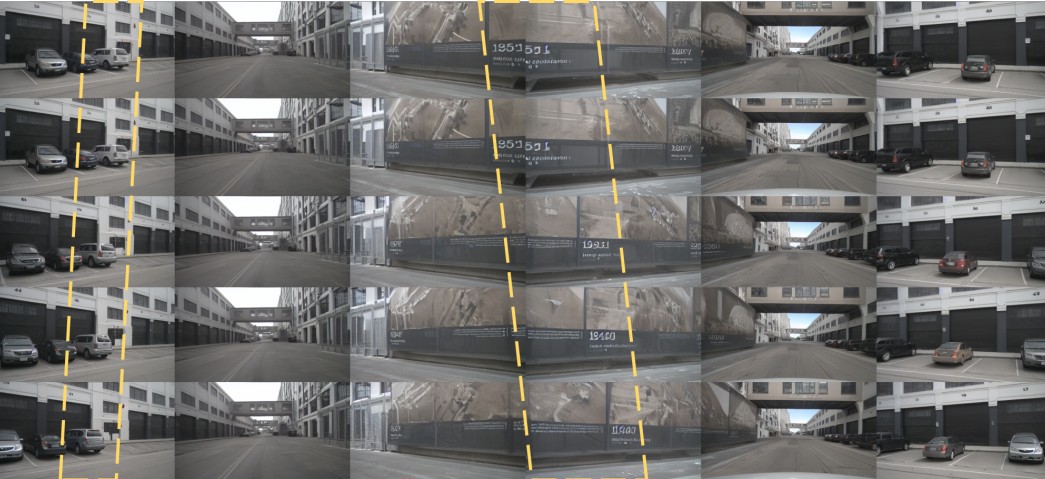

Reverse chronological order

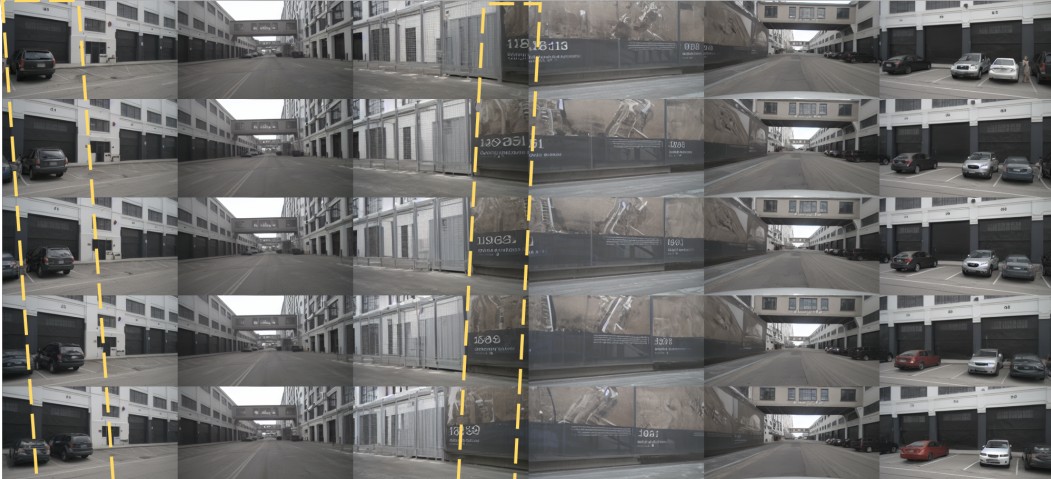

Figure 17: Qualitative results for temporal-level generalization. Our model can generate videos in reverse chronological order to simulate the scene that ego vehicle is moving backward.

Chronological order

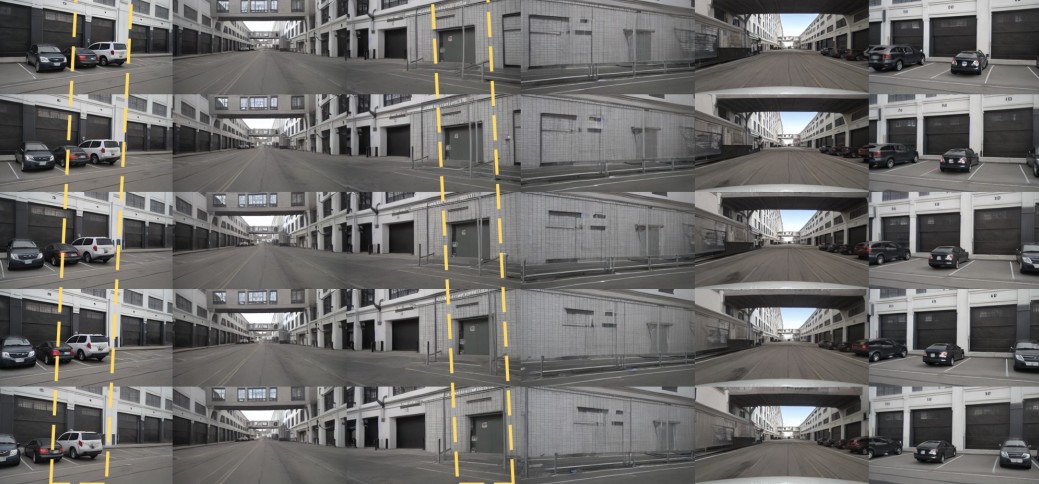

Reverse chronological order

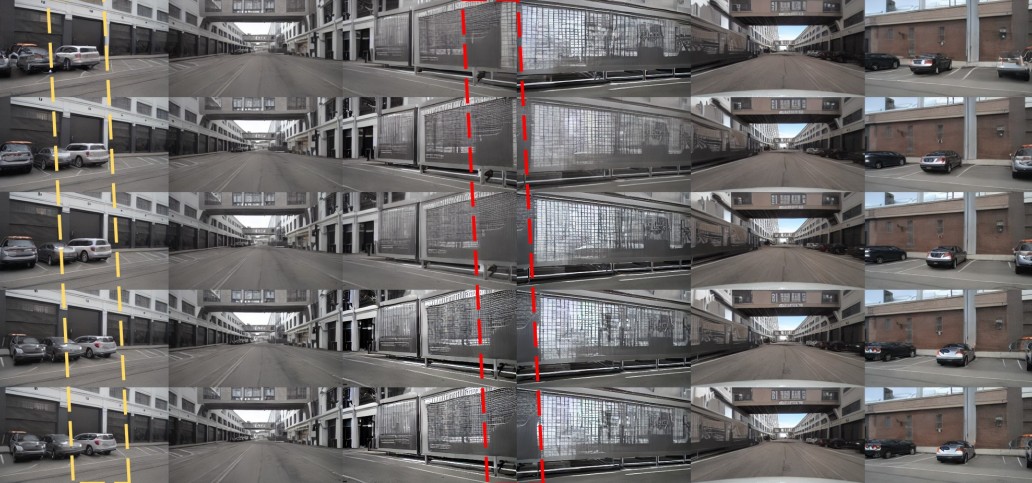

Figure 18: Qualitative results for temporal-level generalization of baseline model DreamForge. When generating in chronological order, the foreground and background should move forward relative to ego vehicle. DreamForge can generate foreground objects correctly due to perspective-based control, but cannot handle background correctly with implicit camera modeling.

