# OpenReview forum: "DriveCamSim: Generalizable Camera Simulation via Explicit Camera Modeling for Autonomous Driving"
_ICLR.cc/2026/Conference — Submitted to ICLR 2026_

### Official Review · Reviewer_BtnN · 2025-10-17

**Soundness:** 1
**Presentation:** 1
**Contribution:** 1
**Rating:** 0
**Confidence:** 5

**Summary:**

This paper focuses on generative model-based camera sensor simulation. The authors think that existing generative methods fail to generalize across different camera viewpoints and video frequencies, and thus propose modules like the Explicit Camera Modeling (ECM) mechanism to achieve Generalizable Camera Simulation based on generative models.

**Strengths:**

The quality of generated views is acceptable. The experiments employ a variety of metrics, including detection (with multiple detectors), segmentation, planning, and occupancy prediction.

**Weaknesses:**

1. I suspect the authors may not fully understand the current problem settings in driving simulation. Generally, current driving simulation problem settings fall into two categories: 1) Imagining a driving scenario (without reference images, e.g., BEVControl); 2) Driving simulation based on one or multiple real images (with reference images, requiring the generated output to align with the real scene, at least for areas visible in the reference frames. This includes most driving world models and Novel View Synthesis (NVS) methods ).
While this paper adopts the inputs of the second setting (using reference images), it completely fails to meet the respective performance requirements. As clearly shown in Figures 1, 5, 6, and 9, the proposed method cannot recover the real scene faithfully, unlike Bench2Drive-R or NVS methods. Therefore, the experimental comparison with Bench2Drive-R is invalid. The authors use Figures 6-12 to demonstrate the NVS capability of their method, but these figures conversely highlight its lack of NVS capability, particularly for foreground objects, which this method specifically emphasize.

2. The authors did not explain how video generation is performed in the absence of real images: Is the first frame generated based on other conditions except reference images, with subsequent frames generated based on previously generated frames? It is hoped that the authors did not use real images as historical/start frames, as this would violate the input requirements of the first problem setting. How can ground truth images be obtained when imagining a driving scenario from scratch?

3. The authors appear unfamiliar with methods under the second problem setting mentioned above (e.g., driving world models based on first-frame real image; NVS methods like SGD, FreeVS, StreetCrafter; or reconstruction-based methods like FreeSim, DrivingGaussian, StreetGaussian). The absence of references to and discussion of these relevant works is likely a main reason for the paper's misunderstanding of the task requirements.

4. Poor language quality. Some sentences are even grammatically incorrect. For example:
"To summary, our contributions as summarized as follows"
“Despite demonstrating impressive performance on standardized benchmarks, critical limitations persist in terms of generalization capability and performance in corner cases.”
"Among these, two representative technical approaches are rendering-based methods and generative models,"
“Performance of UniAD’s different tasks on nuScenes validation set”
The authors may utilize LLMs to improve the language.

**Questions:**

See weakness.

---

> ### Author Response · Authors · 2025-11-26
>
> We sincerely thank the reviewer for providing thoughtful review and constructive suggestions! We kindly ask the reviewer to let us know if further clarification is needed.
>
> > **Q1.1:** I suspect the authors may not fully understand the current problem settings in driving simulation.
>
> **A1.1:**  We fully understand what we are doing in this work, and our motivation is clearly stated in the Introduction.
>
> > **Q1.2:**  Generally, current driving simulation problem settings fall into two categories: 1) Imagining a driving scenario (without reference images, e.g., BEVControl); 2) Driving simulation based on one or multiple real images (with reference images, requiring the generated output to align with the real scene, at least for areas visible in the reference frames. This includes most driving world models and Novel View Synthesis (NVS) methods).
>
> **A1.2:**  We believe your categorization is neither comprehensive nor representative.
>
> 1.	Not comprehensive. According to your categorization, the first category does not have reference images, the second category use real images as reference images, however, there are some methods, which use generated images as reference images, do not belong to any category. These methods include DreamForge, which first uses a single frame model to generate initial frame, then uses this as reference image to autoregressively generate videos.
>
> 2.	Not representative. We believe it is not essential to categorize based on whether reference images or real images are used or not. (a) Use reference images or not: Current video generation methods typically generate a short video clip, due to the limit of computation cost, then use the last frame as reference to autoregressively generate long videos, according to your categorization, this model belongs to the first category when generating the short video clip, but change to another after generating long video. (b) Use real images or not: if the generated image is sufficiently authentic, what is the difference between the model using generated image as reference and the model using real images as reference?
>
> On the contrary, we think our categorization, that categorizes current methods into generative models (learn data distribution from large amount of data) and rendering-based methods (first reconstruct the scene, then render views), is more clear and representative, and better distinguish the advantages and disadvantages of different methods.
>
> > **Q1.3:** While this paper adopts the inputs of the second setting (using reference images)…
>
> **A1.3:** When comparing to other methods in Table.1 and Table.2, we adopt two settings for fair comparison: (a) using 3 previously generated frames as reference, (b) using 1 generated frame and 2 real frames as reference. Based on your definition, our method simultaneously falls into your second category and does not fall into your second category, which makes us really confused.
>
> > **Q1.4:** …, it completely fails to meet the respective performance requirements.
>
> **A1.4:** When comparing to Bench2Drive-R, we adopt the same setting, and achieve better performance, we do not understand why our method “completely fails to meet the respective performance requirements”.
>
> > **Q1.5:** As clearly shown in Figures 1, 5, 6, and 9, the proposed method cannot recover the real scene faithfully, unlike Bench2Drive-R or NVS methods.
>
> **A1.5:** First, we admit the visualization quality is not as good as rendering-based methods. This is determined by the fundamental principles of technology. Rendering-based methods, reconstruct the driving scene with all collected observation and perform offline optimization. Under small viewpoint perturbation, they do have better view consistency and better align with the real scene. However, rendering-based methods often suffer from significant degradation in rendering quality on novel trajectories with large lateral displacements, due to the sparse observation and low construction quality in driving scenes, so we believe generative methods exhibit a higher performance ceiling, that’s why we choose generative method as our start point. For now, it is natural for generative models to lag behind rendering-based methods on visualization quality, under the circumstance of small viewpoint perturbation.
> Second, in the visualization of Bench2Drive-R paper, there are some obvious artifacts. Take Fig.16 as an example, the shape and color of cars change over time, so we do not understand why our method “cannot recover the real scene faithfully” but Bench2Drive-R can.
>
> > **Q1.6:** Therefore, the experimental comparison with Bench2Drive-R is invalid.
>
> **A1.6:** As we said in A1.4, we compare with Bench2Drive-R under same setting, so we do not understand why the experimental comparison is invalid.

---

> ### Author Response · Authors · 2025-11-26
>
> > **Q1.7:** The authors use Figures 6-12 to demonstrate the NVS capability of their method, but these figures conversely highlight its lack of NVS capability, particularly for foreground objects, which this method specifically emphasize.
>
> **A1.7:** As we said in A1.5, we choose generative method as our start point because we think they have a higher performance ceiling, but current research lacks NVS capability as the rendering-based method to generate multi-view videos with novel camera parameters. So our goal is to combine the strength of both branch. To the best of our knowledge, our work is the first to solve this problem, to assign the generative model NVS capability in the field of driving simulation. In the qualitative comparison in Fig.5, when applying camera rotation, our method generate visually correct foreground and background, while previous methods fail. We think it is a strong evidence to show our method has the NVS capability while previous methods do not.
>
> > **Q2.1:** The authors did not explain how video generation is performed in the absence of real images: Is the first frame generated based on other conditions except reference images, with subsequent frames generated based on previously generated frames? It is hoped that the authors did not use real images as historical/start frames, as this would violate the input requirements of the first problem setting.
>
> **A2.1:** For the setting without real images, it is correct that "the first frame generated based on other conditions except reference images, with subsequent frames generated based on previously generated frames". For the setting with real images, the first frame use 3 real frames as reference, while later frames use 2 real frames and 1 generated frame as reference.
>
> > **Q2.2:** How can ground truth images be obtained when imagining a driving scenario from scratch?
>
> **Q2.2:** The classic metrics for generative models, such as FID and FVD, do not need one specific ground truth image for each generated image. Instead, they measure the distance between the feature distributions of generated images and the ground truth dataset.
>
> > **Q3:** The authors appear unfamiliar with methods under the second problem setting mentioned above (e.g., driving world models based on first-frame real image; NVS methods like SGD, FreeVS, StreetCrafter; or reconstruction-based methods like FreeSim, DrivingGaussian, StreetGaussian). The absence of references to and discussion of these relevant works is likely a main reason for the paper's misunderstanding of the task requirements.
>
> **A3:** We believe you have some misunderstanding about what our work is about. Most of the paper you listed are trying to solve the problem of rendering quality degradation for novel ego pose in the single view setting, which is a main problem for rendering-based method, as we said before. Our work, is not to get NVS capability for ego pose, but for novel camera pose in multi-view video generation. For instance, we have vehicles with different sensor configurations, our goal is one generative model could generate multi-view videos with different camera parameters that are suitable for all vehicles.
>
> > **Q4:** Poor language quality. Some sentences are even grammatically incorrect.
>
> **A4:** We thank you for pointing out the typos in paper. We have fixed the typos in revised paper.

---

> ### Comment · Reviewer_BtnN · 2025-11-27
>
> 1. My classification of driving simulation is based on the practical purpose of the method:
> a. If we aim to imagine a scene, there should be no reference image input. At least, it should be an optional input.
> b. If we aim to perform novel view synthesis or expand a scene based on existing observations, a reference image input should be provided, and the generated result should be loyal to the existing scene contents in the originally visible areas.
> When recovering the original scene is not considered, what practical significance does requiring a reference image input have? This significantly reduces the method's practicality compared to approaches that do not require a reference image. If the goal is to generate new scenes with similar structural layouts, this can be more conveniently controlled through abstract scene layout conditions. Even in discussions purely about video generation, I have never encountered a video generation method that do not concern consistency between the generated content and the reference frame.
> 2. I completely disagree that the severe loss of foreground object appearance control demonstrated in the paper is a reasonable disadvantage of generative models. Works such as FreeVS, StreetCrafter and DriveDreamer4D have fully proven that generative models can faithfully recover 3D scenes.
> 3.	The statements “To the best of our knowledge, our work is the **first** to solve this problem, to assign the generative model NVS capability in the field of driving simulation” and “In the qualitative comparison in Fig.5, when applying camera rotation, our method generate visually correct foreground and background, **while previous methods fail**” are entirely wrong. I am surprised that the authors have still not acknowledged the performance and contributions of works like FreeVS, StreetCrafter, FreeSim, and DriveDreamer4D. For example, check Scene.3/Scene.4/Scene.6 on https://freevs24.github.io/. Clearly, prior works have already addressed Generative-model-based NVS, including camera rotation, and they can recover object appearances in reference images with considerable accuracy. Methods like StreetCrafter, FreeSim, and DriveDreamer4D, which incorporate generative models, also possess similar capabilities.
> 4. The authors’ emphasis on distinguishing NVS from novel camera pose deeply confuses me. Changing the camera extrinsic parameters constitutes NVS(novel view synthesis)—there is no difference between the two. Altering the camera intrinsic parameters is generally not discussed in NVS task, but for reconstruction-based methods based on 3DGS, modifying the intrinsic parameters is significantly simpler than changing the extrinsic parameters (as long as the expanded field of view does not exceed the visible part in the original video), as it only involves enlarging the viewing frustum during rendering without changing the viewpoint.
> 5. I reiterate that the NVS task requires the method to recover the content of the scene visible in the reference images, **including foreground objects**. The uncontrolled foreground object appearance demonstrated in figures such as Fig. 7 and Fig.10 is unacceptable for NVS.

---

> > ### Comment · Reviewer_BtnN · 2025-11-27
> >
> > I also checked the newly uploaded supplementary video.  The appearance of foreground objects drastically changes every few frames.  Even if we do not discuss the requirements of NVS task, such temporal consistency is still poor for a video generation method.

---

> > > ### Author Response · Authors · 2025-11-28
> > >
> > > > **Q:** My classification of driving simulation is based on the practical purpose of the method: a. If we aim to imagine a scene, there should be no reference image input. At least, it should be an optional input. b. If we aim to perform novel view synthesis or expand a scene based on existing observations, a reference image input should be provided, and the generated result should be loyal to the existing scene contents in the originally visible areas. When recovering the original scene is not considered, what practical significance does requiring a reference image input have? This significantly reduces the method's practicality compared to approaches that do not require a reference image. If the goal is to generate new scenes with similar structural layouts, this can be more conveniently controlled through abstract scene layout conditions. Even in discussions purely about video generation, I have never encountered a video generation method that do not concern consistency between the generated content and the reference frame.
> > >
> > > **A:** We respectfully insist that your categorization is not representative.
> > >
> > > a. To generate long videos under computational constraints, the standard pipeline first synthesizes an initial frame (or a brief clip) and then employs it as a reference for autoregressive extrapolation; consequently, imaging a scene does not conflict with conditioning on a reference frame.
> > >
> > > b. Consider an intersection where the logged trajectory corresponds to a right turn, yet the simulation commands a left turn: after a short horizon the simulated view no longer overlaps with any real reference image. In this regime, what is the essential distinction between imagining the scene and using the real frame as reference?
> > >
> > >
> > > > **Q:** I completely disagree that the severe loss of foreground object appearance control demonstrated in the paper is a reasonable disadvantage of generative models. Works such as FreeVS, StreetCrafter and DriveDreamer4D have fully proven that generative models can faithfully recover 3D scenes.
> > >
> > > **A:** We appreciate the merits of these works. We want to first clarify that our goal is to build a **scalable**, **reactive**, **multi-view** simulator with **pure generative model**, yet their problem settings and methodological choices diverge from ours in three critical dimensions.
> > >
> > > 1. **Problem scope.**
> > >    FreeVS, StreetCrafter, and DriveDreamer4D address novel ego-pose synthesis in a **single-view** regime; multi-view imagery is obtained by repeatedly forwarding the same model. In contrast, we target **simultaneous multi-view** generation under arbitrary camera rigs, which imposes an additional cross-view consistency constraint that must be satisfied in a single forward pass.
> > >
> > > 2. **Methodological trade-offs.**
> > >    - DriveDreamer4D first generates a video clip and then reconstructs the scene with 4D Gaussian Splatting. The subsequent differentiable rendering stage yields excellent temporal coherence, yet the optimization is performed **offline** and not **pure generative**.
> > >    - StreetCrafter and FreeVS lift colored point clouds into the target view via depth-guided warping. The availability of metric depth substantially simplifies the geometric reasoning task, but restricts the methods to **logged sequences** that contain synchronized LiDAR.
> > >    Our pipeline consumes **only images**, enabling training on massive, low-cost image collections and permitting seamless switching between “imagining” a new scene or “recovering” an existing one.
> > >
> > > 3. **Evaluation fairness.**
> > >    Under the **exact same** setting —purely generative, autoregressive, multi-view, and image-centric—the only directly comparable baseline is Bench2Drive-R. Quantitatively, our approach achieves better performance. Qualitatively, a direct comparison is infeasible because Bench2Drive-R’s code is not publicly available; nevertheless, the examples released in the paper also display noticeable artifacts. We therefore regard the observed degradation in foreground appearance control as an expected limitation of the autoregressive generative paradigm, wherein unidirectional attention allows each frame to interact only with its predecessors, yielding temporally less coherent results than bidirectional video clip generation models.
> > >
> > > We agree that a systematic comparison with the aforementioned works will improve the completeness of our manuscript. A detailed discussion of the methodological distinctions and trade-offs will be added in paper before the discussion period closes. We thank the reviewer for this constructive suggestion.

---

> > ### Author Response · Authors · 2025-11-28
> >
> > > **Q:** The statements “To the best of our knowledge, our work is the first to solve this problem, to assign the generative model NVS capability in the field of driving simulation” and “In the qualitative comparison in Fig.5, when applying camera rotation, our method generate visually correct foreground and background, while previous methods fail” are entirely wrong. I am surprised that the authors have still not acknowledged the performance and contributions of works like FreeVS, StreetCrafter, FreeSim, and DriveDreamer4D. For example, check Scene.3/Scene.4/Scene.6 on https://freevs24.github.io/. Clearly, prior works have already addressed Generative-model-based NVS, including camera rotation, and they can recover object appearances in reference images with considerable accuracy. Methods like StreetCrafter, FreeSim, and DriveDreamer4D, which incorporate generative models, also possess similar capabilities.
> >
> > **A:** We respectfully reiterate the distinction articulated above: our work targets the simultaneous synthesis of multi-view imagery under arbitrary inter-camera relative poses, whereas the works you mentioned address novel ego-pose synthesis in a single-view regime. We fully acknowledge the impressive capabilities of the cited works and will expand the revised manuscript with a detailed discussion that clarifies this methodological gap and properly credits their contributions.
> >
> > > **Q:** The authors’ emphasis on distinguishing NVS from novel camera pose deeply confuses me. Changing the camera extrinsic parameters constitutes NVS(novel view synthesis)—there is no difference between the two. Altering the camera intrinsic parameters is generally not discussed in NVS task, but for reconstruction-based methods based on 3DGS, modifying the intrinsic parameters is significantly simpler than changing the extrinsic parameters (as long as the expanded field of view does not exceed the visible part in the original video), as it only involves enlarging the viewing frustum during rendering without changing the viewpoint.
> >
> > **A:** We believe the preceding response addresses your question about camera extrinsics. Regarding intrinsics, we acknowledge that rendering-based approaches can, in principle, adjust focal length; however, our method operates within the purely generative paradigm, where no prior work under same setting has demonstrated controllable intrinsic editing. Moreover, Figure 13 illustrates that our model not only increases focal length to zoom in, but also decreases it to zoom out, synthesizing semantically consistent content beyond the original image borders, thus establishing it as a superior alternative to zero-padding. When the expanded field of view exceeds the visible portion of the source video, such extrapolation remains beyond the capability of standard rendering-based techniques.
> >
> > > **Q:** I reiterate that the NVS task requires the method to recover the content of the scene visible in the reference images, including foreground objects. The uncontrolled foreground object appearance demonstrated in figures such as Fig. 7 and Fig.10 is unacceptable for NVS. I also checked the newly uploaded supplementary video. The appearance of foreground objects drastically changes every few frames. Even if we do not discuss the requirements of NVS task, such temporal consistency is still poor for a video generation method.
> >
> > **A:** We reiterate that we target scalable, reactive, closed-loop simulation, where the next camera pose is unknown in advance and is produced on-the-fly by the AD model. To this end we follow the  protocol introduced in Bench2Drive-R. Compared with the methods that generate whole video clip or introducing extra lidar information, this protocol inevitably yields larger photometric variance.
> >
> > As Bench2Drive-R’s paper likewise shows foreground flicker, we believe that the issue is inherent to the autoregressive paradigm rather than a method-specific failure. We nevertheless retain this paradigm because it is necessary that allows the simulator to react instantaneously to the AD policy’s trajectory output, which is essential for closed-loop evaluation. Improving temporal consistency while preserving reactivity is an ongoing research direction, and we will discuss this limitation and its root cause in the revised manuscript.

---

### Official Review · Reviewer_5LVh · 2025-10-25

**Soundness:** 3
**Presentation:** 3
**Contribution:** 4
**Rating:** 6
**Confidence:** 5

**Summary:**

This paper presents DriveCamSim, a generalizable camera simulation framework for autonomous driving, addressing the limitations of existing methods in fixed camera configurations and video frequencies. Its core innovations include Explicit Camera Modeling (ECM) for cross-view and cross-frame interactions in 3D physical space, an information-preserving control mechanism to enhance controllability and temporal consistency, and supporting strategies like overlap-based view matching and random frame sampling. Extensive experiments on nuScenes and nuPlan datasets demonstrate superior performance in visual quality, controllability, and generalization across spatial and temporal levels.

**Strengths:**

1. The proposed ECM mechanism effectively decouples the model from specific camera parameters and temporal sampling rates by establishing explicit pixel-wise correspondences in 3D space, filling the gap of poor generalization in existing implicit modeling methods.
2. The information-preserving control mechanism, especially the identity-aware extension, successfully mitigates information loss in conditional encoding and injection, improving both controllability and foreground temporal consistency.
3. The design of overlap-based view matching and random frame sampling strategies complements ECM well, enhancing the model’s ability to utilize relevant context and avoid over-reliance on adjacent frames.
4. The framework supports flexible user-customizable simulation (e.g., variable frame rates, reverse temporal order), showing strong practical value for downstream autonomous driving algorithm evaluation and data augmentation.

**Weaknesses:**

1. Using MagicDrive and DreamForge as baselines is insufficient, as they do not support camera parameter generalization. On the contrary, the paper should conduct a direct comparison with 3D-based generative works like MagicDrive3D[a] to show advantages.
2. No video-specific evaluation metrics are employed. Benchmarks like W-CODA2024[b], which are tailored for video generation quality and consistency, should be adopted to comprehensively assess temporal performance.
3. Key components in Figure 4, Table 5, and Section 3.4 lack sufficient references to previous works. For example, "Text Condition" and "3D Bounding Boxes Encoding" draw on MagicDrive, and "Perspective-based control" originates from BEVControl, but these connections are not clearly cited.
4. The experimental section does not provide training support for downstream perception tasks. It remains unclear how the generated data is integrated into the training pipeline of models like BEVFusion and what advantage this method can bring.
5. The model struggles with large camera parameter perturbations (e.g., significant translation/rotation in x/z axes) and lags behind rendering-based methods in view consistency, indicating room for improvement in sensor modeling.

[a] https://arxiv.org/abs/2405.14475

[b] https://arxiv.org/abs/2507.01735

**Questions:**

1. For the Explicit Camera Modeling (ECM) mechanism, the paper mentions setting several depth anchors to project query view pixels to 3D space. What criteria were used to determine the number (e.g., 10) and range ([1, 60]) of these depth anchors, and how would adjusting these parameters impact the model’s performance in building pixel correspondences?
2. The model is built upon a pre-trained Stable Diffusion v1.5, and all parameters of the UNet are retrained. Why was Stable Diffusion v1.5 chosen as the base model instead of other better models (e.g., Stable Diffusion XL or Pixart) or video diffusion models (e.g., Wan 2.1), and how does the choice of the base model influence the model’s initial performance and training convergence speed?

---

> ### Author Response · Authors · 2025-11-26
>
> We sincerely thank the reviewer for providing thoughtful review and constructive suggestions! We kindly ask the reviewer to let us know if further clarification is needed.
>
> > **Q1:** Using MagicDrive and DreamForge as baselines is insufficient, as they do not support camera parameter generalization. On the contrary, the paper should conduct a direct comparison with 3D-based generative works like MagicDrive3D to show advantages.
>
> **A1:** We want to make a clarification about the difference between our method and MagicDrive3D. MagicDrive3D first uses a video generator to generate a video clip, whose camera parameters still overfit to training dataset, and then reconstructs the scene with 3DGS. While this is a promising two-stage pipeline to ensure the view consistency for any view rendering, it must generate the whole video clip before novel view rendering. However, our ultimate goal is to build a reactive simulator, that is why we adopt auto-regressive image generation rather than whole clip video generation, so we need to support novel view at generation stage. Furthermore, it is unfeasible for us to quantitatively compare with MagicDrive3D on reconstruction metrics like PSNR or SSIM, since we do not have a reconstruction module.
>
> To quantitatively verify how good the generated images follow the input camera parameters, we conduct a camera pose estimiation experiment using generated images. We adopt VGGT [a] as a modern feed-forward SfM model and follow PoseDiffusion [b] to report Relative Rotation Accuracy (RRA) and Relative Translation Accuracy (RTA) at two thresholds (@15, @30). As shown below, ID-3, with rotated camera parameters, attains comparable RRA@30 and better RRA@15 than ID-2. We think this is because the rotation enlarges the overlap between neighbor views, which is small for original camera parameters. To validate this, we further conduct ID-4 to generate images with two additional virtual views, resulting in better RRA. In contrast, RTA appears unreliable in this setting—VGGT yields higher RTA on our generated images than on the real images—so we consider RRA as the primary metric. we think the results provide more evidence about the generalization ability of our method.
>
> | ID | Image Source | Camera Pose | RRA@30(↑) | RTA@30(↑) | RRA@15(↑) | RTA@15(↑) |
> | :---------: | :---------: | :---------: | :---------: | :---------: | :---------: | :---------: |
> | 1 | Real | Original | 0.9089 | 0.4400 | 0.8467 | 0.2049 |
> | 2 | Generated | Original | 0.5187 | 0.4858 | 0.2191 | 0.2622 |
> | 3 | Generated | Rotation | 0.5133 | 0.4329 | 0.2671 | 0.2240 |
> | 4 | Generated | Add virtual views | 0.6010 | 0.3998 | 0.2926 | 0.1957 |
>
>
> > **Q2:** No video-specific evaluation metrics are employed. Benchmarks like W-CODA2024, which are tailored for video generation quality and consistency, should be adopted to comprehensively assess temporal performance.
>
> **A2:** Thanks for your suggestion! We have added FVD metric on W-CODA benchmark as follows. Our method surpasses DreamForge, which ranks second on the benchmark, while the first place solution DiVE adopts a more advanced DiT-based pretrained model.
>
> | Method | FVD |
> | :---------: | :---------: |
> | DiVE | 94.5979 |
> | DreamForge | 224.7638 |
> | DiVE | 195.5768 |
>
>
> > **Q3:** Key components in Figure 4, Table 5, and Section 3.4 lack sufficient references to previous works. For example, "Text Condition" and "3D Bounding Boxes Encoding" draw on MagicDrive, and "Perspective-based control" originates from BEVControl, but these connections are not clearly cited.
>
> **A3:** Thanks for your suggestions! We have added the references in revised paper.
>
> > **Q4:** The experimental section does not provide training support for downstream perception tasks. It remains unclear how the generated data is integrated into the training pipeline of models like BEVFusion and what advantage this method can bring.
>
> **A4:** Thanks for your question! We perform training support experiments on SparseDrive stage-1 as follows. Trained solely on generated videos, SparseDrive achieves 41.03 NDS and 48.14 mAP, only slightly lags behind training on real data. Using synthetic videos for data augmentation can further boost performance to surpass the model trained on real data.
>
> | Real | Generated | detection: NDS | online mapping: mAP |
> | :---------: | :---------: | :---------: | :---------: |
> | √ | | 45.58 | 51.77 |
> |  | √ | 41.03 | 48.14 |
> | √ | √ | 49.13 | 53.65 |

---

> ### Author Response · Authors · 2025-11-26
>
> > **Q5:** The model struggles with large camera parameter perturbations (e.g., significant translation/rotation in x/z axes) and lags behind rendering-based methods in view consistency, indicating room for improvement in sensor modeling.
>
> **A5:** Thanks for pointing out the limitation. We admit there are room for performance improvement, e.g. more training data, larger model. We want to emphasize that this work is a preliminary exploration to combine the advantages of both diffusion-based generative models and rendering-based methods, the quantitative and qualitative results have shown that our method have the potential to achieve this goal. Our method achieves SOTA performance compared with other methods using same pretrained model, and we will upgrade our method with better pretrained model for better performance once we have enough GPU resources.
>
> > **Q6:** For the Explicit Camera Modeling (ECM) mechanism, the paper mentions setting several depth anchors to project query view pixels to 3D space. What criteria were used to determine the number (e.g., 10) and range ([1, 60]) of these depth anchors, and how would adjusting these parameters impact the model’s performance in building pixel correspondences?
>
> **A6** We conducted early ablation experiments to choose the number of depth anchors and different depth range. The final hyperparameters are chosen considering the tradeoff between generation performance and computation cost.
>
>
> > **Q7:** The model is built upon a pre-trained Stable Diffusion v1.5, and all parameters of the UNet are retrained. Why was Stable Diffusion v1.5 chosen as the base model instead of other better models (e.g., Stable Diffusion XL or Pixart) or video diffusion models (e.g., Wan 2.1), and how does the choice of the base model influence the model’s initial performance and training convergence speed?
>
> **A7:** We use Stable Diffusion v1.5 for fair comparison with other classic methods in this field. We think a better base model would lead to better performance, and the convergence speed should be related to both base model and the data distribution. If the base model was ever trained on driving images similar to nuScenes dataset, it should converge faster.
>
> [a] Wang, Jianyuan, et al. "Vggt: Visual geometry grounded transformer." Proceedings of the Computer Vision and Pattern Recognition Conference. 2025.
>
> [b] Wang, Jianyuan, Christian Rupprecht, and David Novotny. "Posediffusion: Solving pose estimation via diffusion-aided bundle adjustment." Proceedings of the IEEE/CVF International Conference on Computer Vision. 2023.

---

### Official Review · Reviewer_Yu4o · 2025-10-30

**Soundness:** 3
**Presentation:** 3
**Contribution:** 2
**Rating:** 4
**Confidence:** 4

**Summary:**

Motivated by the goal of integrating the strengths of both rendering-based techniques and generative models while addressing their respective limitations, this paper proposes a generalizable camera simulation framework, DriveCamSim. The framework uses a novel ECM mechanism to simulate camera sensors. The ECM effectively mitigates overfitting to the specific camera sensors present in the training data. Additionally, for controllable generation, the paper introduces an information-preserving control mechanism that significantly enhances controllability. Through extensive experiments, this work demonstrates state-of-the-art performance across a variety of scenes.

**Strengths:**

- A novel and compact explicit camera modeling mechanism is proposed.
- Detailed visualization results are provided, offering valuable insights.

**Weaknesses:**

- In Table 3, the perspective-based and attention-based control mechanisms are presented, but it is unclear which methods these mechanisms correspond to.
- The novelty of the approach is not immediately apparent in the methods section, as it contains a lot of detailed explanations about handling different conditions.

**Questions:**

- Can the FVD score be included in Table 1?
- From what I understand, the ECM uses intrinsic and extrinsic camera parameters to improve generation robustness. However, how does its performance compare to the native design that directly inputs the camera parameters as conditions?
- As the core contribution of this work, the explicit camera modeling experiment in Table 3 is somewhat unconvincing. Additionally, based on the images presented in the supplement, the generative effect significantly worsens when the rotation exceeds 10°. Could you provide more generative video examples in the supplementary materials?
- Regarding the information-preserving control mechanism, could you provide quantitative and visualization comparisons between the current approach and methods that suffer from critical information loss?

---

> ### Author Response · Authors · 2025-11-26
>
> We sincerely thank the reviewer for providing thoughtful review and constructive suggestions! We kindly ask the reviewer to let us know if further clarification is needed.
> > **Q1:** In Table 3, the perspective-based and attention-based control mechanisms are presented, but it is unclear which methods these mechanisms correspond to.
>
> **A1:** Thanks for your question. The perspective-based control mechanism originates from BEVControl and are employed by most previous work, such as Panacea, DriveDreamer, etc. The attention-based control mechanism is proposed by MagicDrive. We have updated the reference in Table 3 in revised paper.
>
>
> > **Q2:** The novelty of the approach is not immediately apparent in the methods section, as it contains a lot of detailed explanations about handling different conditions.
>
> **A2:** Thanks for your question. The methods section mainly describes the implenmentation details, which is necessary for the readers unfamiliar with this field. The novelty of our method is mainly illustrated in our motivation in Introduction and validated by extensive qualitative and quantitative experiments.
>
>
> > **Q3:** Can the FVD score be included in Table 1?
>
> **A3:** In Table 1, BEVControl and MagicDrive are image generation methods, Bench2Drive-R did not report FVD metric in paper. Panacea, though reported FVD in paper, did not explain how they calculated FVD in details. To compare FVD metric fairly, we add results on W-CODA benchmark [a] as follows. Our method surpasses DreamForge, which ranks second on the benchmark, while the first place solution DiVE adopts a more advanced DiT-based pretrained model. We have added the results in revised paper.
>
>
> | Method | FVD |
> | :---------: | :---------: |
> | DiVE | 94.5979 |
> | DreamForge | 224.7638 |
> | DiVE | 195.5768 |
>
> > **Q4:** From what I understand, the ECM uses intrinsic and extrinsic camera parameters to improve generation robustness. However, how does its performance compare to the native design that directly inputs the camera parameters as conditions?
>
> **A4:** MagicDrive is the representative method that directly inputs the camera parameters as conditions. As shown in Table 1 and Table 2, our method achieves better performance than MagicDrive.

---

> ### Author Response · Authors · 2025-11-26
>
> > **Q5:** As the core contribution of this work, the explicit camera modeling experiment in Table 3 is somewhat unconvincing. Additionally, based on the images presented in the supplement, the generative effect significantly worsens when the rotation exceeds 10°. Could you provide more generative video examples in the supplementary materials?
>
> **A5:** Thanks for your advice. We have uploaded videos as supplementary materials. Additionally, to quantitatively validate if ECM could generate images respecting the input camera parameters, we conduct a camera pose estimiation experiment using generated images. We adopt VGGT [b] as a modern feed-forward SfM model and follow PoseDiffusion [c] to report Relative Rotation Accuracy (RRA) and Relative Translation Accuracy (RTA) at two thresholds (@15, @30). As shown below, ID-3, with rotated camera parameters, attains comparable RRA@30 and better RRA@15 than ID-2. We think this is because the rotation enlarges the overlap between neighbor views, which is small for original camera parameters. To validate this, we further conduct ID-4 to generate images with two additional virtual views, resulting in better RRA. In contrast, RTA appears unreliable in this setting—VGGT yields higher RTA on our generated images than on the real images—so we consider RRA as the primary metric. We think the results provide more evidence about the generalization ability of our method.
>
> | ID | Image Source | Camera Pose | RRA@30(↑) | RTA@30(↑) | RRA@15(↑) | RTA@15(↑) |
> | :---------: | :---------: | :---------: | :---------: | :---------: | :---------: | :---------: |
> | 1 | Real | Original | 0.9089 | 0.4400 | 0.8467 | 0.2049 |
> | 2 | Generated | Original | 0.5187 | 0.4858 | 0.2191 | 0.2622 |
> | 3 | Generated | Rotation | 0.5133 | 0.4329 | 0.2671 | 0.2240 |
> | 4 | Generated | Add virtual views | 0.6010 | 0.3998 | 0.2926 | 0.1957 |
>
> > **Q6:** Regarding the information-preserving control mechanism, could you provide quantitative and visualization comparisons between the current approach and methods that suffer from critical information loss?
>
> **A6:** We previously provided ablation expetiment for different control mechanism in Appendix due to the page limit, and now we have moved ablation part to Sec 4.4 in main paper. As shown in Table 7, our proposed control mechanism achieves better performance than other methods.
>
> Qualitatively, as shown in the scond row of Fig 5, MagicDrive, which uses attention-based control, has no response to camera parameter changes, suffering from relative pose information loss between boxes and the cameras. We further provide visualization to illustrate the information loss for perspective-based control in Appendix D.1. As shown in Fig 6 (a), when we alter the 3D box conditions while keeping the projected 2D boxes identical, the generated foreground always remains the same, suffering from depth information loss. The 3D-to-2D projection process also discards yaw information, resulting in wrong heading for generated vehicles, as shown in Fig 6 (b).
>
>
>
> [a] Chen, Kai, et al. "ECCV 2024 W-CODA: 1st Workshop on Multimodal Perception and Comprehension of Corner Cases in Autonomous Driving." arXiv preprint arXiv:2507.01735 (2025).
>
> [b] Wang, Jianyuan, et al. "Vggt: Visual geometry grounded transformer." Proceedings of the Computer Vision and Pattern Recognition Conference. 2025.
>
> [c] Wang, Jianyuan, Christian Rupprecht, and David Novotny. "Posediffusion: Solving pose estimation via diffusion-aided bundle adjustment." Proceedings of the IEEE/CVF International Conference on Computer Vision. 2023.

---

> > ### Comment · Reviewer_Yu4o · 2025-11-26
> > **Official Comment by Reviewer Yu4o**
> >
> > Thank you for your reply. I believe the necessity of explicit camera modeling for autonomous driving warrants further discussion. Therefore, I would like to maintain my original score.

---

> > > ### Author Response · Authors · 2025-11-27
> > >
> > > Thank you for your careful reading of our response. We hope that our earlier responses have addressed most of your concerns. Regarding your remaining question about the necessity of explicit camera modeling, we would like to offer the following elaboration.
> > >
> > > Our work is motivated by a practical industrial challenge: production vehicles are equipped with heterogeneous camera rigs—different intrinsics, extrinsics, and numbers of sensors. To effectively train and evaluate AD algorithms across such heterogeneous fleets, a generative model must be capable of synthesizing images corresponding to any given camera configuration. However, current simulators face significant limitations:
> > > 1.  **Carla**, while highly popular in academia and configurable for specific camera setups, suffers from limited visual fidelity, introducing a substantial domain gap.
> > > 2.  **Prior generative model approaches** focuse on enhancing visual realism, but inherently assume fixed and unchanging camera parameters – an assumption that does not hold in real-world scenarios. Consequently, these models learn in 2D pixel space with implicit camera modeling, lacking 3D spatial understanding and can generate only images matching the specific camera parameters used during their training.
> > >
> > > While training a dedicated generative model for every possible camera configuration is a potential workaround, this solution fails to leverage the scaling benefits of large amount of data and cannot support the development of downstream models that require novel viewpoints.
> > >
> > > In contrast, our explicit camera modeling provides a promising solution to serve as a foundation generative model that scales across diverse data resources. Table 4 already provides empirical evidence: joint training on nuPlan and nuScenes (different rigs) improves performance over single-dataset training, indicating that the network is leveraging shared 3D scene knowledge rather than memorizing viewpoint-specific patterns. Furthermore, when a new vehicle arrives—whose cameras may never have been seen during training—our model can still generate photo-realistic imagery without additional fine-tuning, facilitating downstream tasks. We believe this capability is a prerequisite for a truly generalizable simulation framework for autonomous driving.
> > >
> > > We hope this clarification underscores the necessity of explicit camera modeling and could resolve your remaining concern. We want to thank you for your time again and look forward to your feedback on our responses.

---

> > > > ### Comment · Reviewer_Yu4o · 2025-11-27
> > > > **Official Comment by Reviewer Yu4o**
> > > >
> > > > Thank you for your prompt response. Regarding your statement that the cameras 'may never have been seen during training,' could you provide both quantitative and qualitative experimental evidence to support this claim? Based on my understanding, I have strong reservations about its validity.

---

> > > > > ### Author Response · Authors · 2025-11-27
> > > > >
> > > > > Thanks for your question! In nuScenes dataset, images are collected using two cars with **identical sensor layout**, as described in the **Car Setup** part of nuScenes Paper [a], so the camera pose of six views are fixed, and this is the only camera configuration that have been seen during training. When we apply translation and rotation to the camera extrinsic, scale the focal length of camera intrinsic and insert virtual camera views, they are all unseen during training. Qualitatively, as shown in Figure 1 and Figure 5, with these unseen camera parameters as input, our method can generate visually correct foreground and background, but previous methods fail to response to the change of camera setup. Quantitatively, we have added new experiment in response A5, that images generated with rotated cameras achieve comparable or even better performance on the camera pose estimation task, compared with images generated with original camera parameters. This demonstrates that the generated images follow the input camera parameters. We hope this could answer your question.
> > > > >
> > > > > [a] https://arxiv.org/pdf/1903.11027

---

### Official Review · Reviewer_MJyW · 2025-10-31

**Soundness:** 3
**Presentation:** 3
**Contribution:** 3
**Rating:** 6
**Confidence:** 4

**Summary:**

This paper presents DriveCamSim, a generalizable camera simulation framework for autonomous driving (AD) that aims to overcome the limitations of existing generative models. The core innovation of DriveCamSim is the Explicit Camera Modeling (ECM) mechanism. Instead of relying on standard attention, ECM establishes explicit pixel-wise correspondences across multiple views and frames, effectively decoupling the model from the fixed spatial and temporal parameters of the training set.
Furthermore, the paper addresses the problem of information loss in conditional generation by proposing an information-preserving control mechanism. This design not only improves the fidelity of controllable generation but is also extended to be identity-aware, enhancing the temporal consistency of foreground objects like vehicles.

**Strengths:**

1. The proposed Explicit Camera Modeling (ECM) is a strong technical contribution that directly addresses a major limitation in prior work. Decoupling the model from fixed camera configurations is a well-motivated and crucial step toward creating truly flexible and practical AD simulators. And the qualitative result looks great.

2. Identity-aware embedding is inspiring, maintaining consistency for dynamic objects, which is a common failure point in generative models.

3. The qualitative and quantitative experiment results show that the proposed designs yield good performance.

**Weaknesses:**

1. The proposed Explicit Camera Modeling enables the model to generalize to unseen parameters. Although there are qualitative experiments showing that the model outperforms previous methods in the generalization of camera configuration. It would be better if there are quantitative evaluations with modern feed-forward SFM models, showing that the generated image follows the desired camera parameters.

2. Ablation is an important part and should be included in the main paper.

3. The submission doesn't include video results, which makes me a bit concerned about the temporal consistency.

**Questions:**

1. What is the number of keypoints per bounding box in the current setting? How are the keypoints sampled? Do the authors think adding the keypoint numbers or doing dense feature matching will improve the generation/identity preserving ability?

---

> ### Author Response · Authors · 2025-11-26
>
> We sincerely thank the reviewer for providing thoughtful review and constructive suggestions! We kindly ask the reviewer to let us know if further clarification is needed.
> > **Q1:** It would be better if there are quantitative evaluations with modern feed-forward SFM models, showing that the generated image follows the desired camera parameters.
>
> **A1:**  Thank you for the suggestion! To quantitatively verify how good the generated images follow the input camera parameters, we conduct a camera pose estimiation experiment using generated images. We adopt VGGT [a] as a modern feed-forward SfM model and follow PoseDiffusion [b] to report Relative Rotation Accuracy (RRA) and Relative Translation Accuracy (RTA) at two thresholds (@15, @30). As shown below, ID-3, with rotated camera parameters, attains comparable RRA@30 and better RRA@15 than ID-2. We think this is because the rotation enlarges the overlap between neighbor views, which is small for original camera parameters. To validate this, we further conduct ID-4 to generate images with two additional virtual views, resulting in better RRA. In contrast, RTA appears unreliable in this setting—VGGT yields higher RTA on our generated images than on the real images—so we consider RRA as the primary metric. we think the results provide more evidence about the generalization ability of our method.
>
> | ID | Image Source | Camera Pose | RRA@30(↑) | RTA@30(↑) | RRA@15(↑) | RTA@15(↑) |
> | :---------: | :---------: | :---------: | :---------: | :---------: | :---------: | :---------: |
> | 1 | Real | Original | 0.9089 | 0.4400 | 0.8467 | 0.2049 |
> | 2 | Generated | Original | 0.5187 | 0.4858 | 0.2191 | 0.2622 |
> | 3 | Generated | Rotation | 0.5133 | 0.4329 | 0.2671 | 0.2240 |
> | 4 | Generated | Add virtual views | 0.6010 | 0.3998 | 0.2926 | 0.1957 |
>
> > **Q2:** Ablation is an important part and should be included in the main paper.
>
> **A2:** Thanks for your advice! The ablation experiments are included into Appendix due to the page limit in submitted version, we have moved the ablation part into the main paper (Sec 4.4) in revised version.
>
> > **Q3:** The submission doesn't include video results, which makes me a bit concerned about the temporal consistency.
>
> **A3:** For your concern about temporal consistency, we provide more quantitative and qualitative results. Quantitatively, we provide FVD metric on W-CODA benchmark [c] as follows. Our method surpasses DreamForge, which ranks second on the benchmark, while the first place solution DiVE adopts a more advanced DiT-based pretrained model.  Qualitatively, we have uploaded videos as supplementary materials.
>
> | Method | FVD |
> | :---------: | :---------: |
> | DiVE | 94.5979 |
> | DreamForge | 224.7638 |
> | DiVE | 195.5768 |
>
>
> > **Q4:** What is the number of keypoints per bounding box in the current setting? How are the keypoints sampled? Do the authors think adding the keypoint numbers or doing dense feature matching will improve the generation/identity preserving ability?
>
> **A4:** We set three fixed values in the x, y, and z directions of the bounding box, and combine them to obtain 27 fixed keypoints distributed around the bounding box. In addition, we set 8 learnable keypoints, resulting in a total of 35 keypoints.
> We do find in some cases, e.g. surrounding vehicles are too close to ego vehicle thus are partially visible in images, some keypoints will project outside the images, resulting in insufficient foreground feature. We have also tried more keypoints (e.g. 60 keypoints), yielding similar generation metrics. We think your suggestion about dense feature matching have the potential to resolve this problem, and leave this for future work.
>
>
> [a] Wang, Jianyuan, et al. "Vggt: Visual geometry grounded transformer." Proceedings of the Computer Vision and Pattern Recognition Conference. 2025.
>
> [b] Wang, Jianyuan, Christian Rupprecht, and David Novotny. "Posediffusion: Solving pose estimation via diffusion-aided bundle adjustment." Proceedings of the IEEE/CVF International Conference on Computer Vision. 2023.
>
> [c] Chen, Kai, et al. "ECCV 2024 W-CODA: 1st Workshop on Multimodal Perception and Comprehension of Corner Cases in Autonomous Driving." arXiv preprint arXiv:2507.01735 (2025).

---

### Meta-Review · Area_Chair_bjxW · 2025-12-30

**Summary:**

Reviewers' concerns primarily focus on:
1. Insufficient comparison with advanced works. The baselines are limited, lacking comparative experiments with state-of-the-art NVS methods(e.g. FreeVS, FreeSim.)
2. Video quality evaluation. The supplementary results still demonstrate poor FVD scores, indicating a need for proper video quality assessment.
3. Generation quality degradation under large camera perturbations.

**Reviewer Concerns:**

Althoug issues such as citation omissions, ablation studies, and camera angle accuracy evaluations have been addressed, the generation quality under large viewpoint shifts has not been solved, a direct comparison with advanced NVS methods is still missing, and the viewpoint consistency of the quantitative results remains poor.

**Reviewer Scores:**

The reviewers MJyW may raise their scores after the discussion, since most of their concerns are addressed, except for the computation cost.

---

### Decision · Program_Chairs · 2026-01-26

Reject